# New advances in jellyfish anatomy: the benefits of endocasts and X-ray microtomography in the investigation of the gastrovascular system of *Cotylorhiza tuberculata* (Scyphozoa; Rhizostomeae; Cepheidae)

Gregorio Motta[1,2]*, Marco Voltolini[3], Lucia Mancini[4], Diego Dreossi[5], Francesco Brun[6], Valentina Tirelli[7,8], Lorenzo Peter Castelletto[1], Manja Rogelja[9], Antonio Terlizzi[1,8], Massimo Avian[1]

1 Department of Life Sciences, University of Trieste, Trieste, Italy, 2 Department of Integrative Marine Ecology, Stazione Zoologica Anton Dohrn, Napoli, Italy, 3 Department of Earth Science Ardito Desio, University of Milano, Milano, Italy, 4 Slovenian National Building and Civil Engineering Institute, Ljubljana, Slovenia, 5 Elettra-Sincrotrone Trieste S.C.p.A., Basovizza, Trieste, Italy, 6 Department of Engineering and Architecture, University of Trieste, Trieste, Italy, 7 National Institute of Oceanography and Applied Geophysics, Trieste, Italy, 8 National Biodiversity Future Center (NBFC), National Biodiversity Future Center, Palermo, Italy, 9 University of Primorska, Aquarium Piran, Piran, Slovenia

* gregorio.motta@units.it

## Abstract

Historically, research on jellyfish anatomy has been viewed as secondary in importance and has not benefited from technical advances that could improve the quality of the results when compared to other disciplines. The most notable example is the anatomical research on jellyfish, which has been done using conventional methods for many years. Thus far, recent studies have shown that X-ray microtomography (µCT) and resin endocasts can yield outputs with remarkably high detail quality. The application of a similar protocol to *Cotylorhiza tuberculata* has allowed us to redescribe the anatomy of this species' gastrovascular system, providing numerous additional details, among them the double constricted canal structure present in the oral arms, which was absent in previous descriptions. Additionally, functional anatomy experiments have revealed a double circulation system within these canals, featuring specialized oral arms' openings for intake and outflow, as previously observed in *Rhizostoma pulmo*. These findings challenge the theory of a simple digestive system in scyphozoans featuring openings that acts both as mouths and anuses. Given the genetic distance between *Cotylorhiza tuberculata* and *Rhizostoma pulmo*, which belong to different suborders (Kolpophorae and Dactyliophorae, respectively), we propose that this complex gastrovascular circulation pattern may be more widespread among the Rhizostomeae.

**Data availability statement:** All relevant data are within the paper and it's Supporting Information files.

**Funding:** This work was partially funded by the National Biodiversity Future Centre (NBFC) Program, Italian Ministry of University and Research, PNRR, Missione 4 Componente 2 Investimento 1.4 (Project: CN00000033) to A.T. The funder had no role in study design, data collection and analysis, decision to publish, or preparation of the manuscript.

**Competing interests:** The authors have declared that no competing interests exist.

## Introduction

Medusozoa (phylum Cnidaria) is one of the oldest group of metazoans, whose fossils date back over 500 million years [1]. Despite their long evolutionary history, they are still relatively understudied compared to other marine organisms [2,3]. The first scientific studies on jellyfish can be traced back to the 18th and 19th centuries and focused on taxonomy and basic morphology [4–6]. In recent decades, a new wave of studies on their physiology, behavior, and ecological roles has emerged due to their increasing abundance and impact on ecosystems and human activities [2,3,7–11]. Despite this recent increase in scientific interest, general knowledge of jellyfish remains limited compared to other taxonomic groups, and many aspects of their biology and ecology are still poorly understood [12,13]. Numerous jellyfish species have received limited scientific attention and the description of their anatomy and physiology are largely undocumented; moreover, much of the existing knowledge is based on a few well-known species often based on old-fashioned research. This state of jellyfish research collides with other scientific fields where studies and references are continually being updated [12].

Jellyfish zoology may be one of the few research areas where updates to techniques and instruments have not been adequately considered, missing the opportunity to make significant advances [14]. Such updates could help address these persistent limitations and provide new and more detailed descriptions that can support or challenge the validity of older reports.

Most of research now focus on the "external" functioning of jellyfish as how their presence impact ecosystems or how oceanographic conditions affect jellyfish abundance and distributions [8–10,15–17]. Despite this knowledge is expanding, a proper description of "internal" anatomy and physiology is missing for most of the species, and this is one of the most important deficiencies to address. New technologies could help overcome historical problems in anatomical and zoological studies, as the opacity and thickness of the mesoglea layer severely compromising the quality and accuracy of traditional observations. Advances in anatomical studies such that of Avian and colleagues [18], who proposed a brand-new use of endocasts and microtomography for the three-dimensional analysis of the gastrovascular system of jellyfish, have significantly overtaken traditional analyses and the classic descriptions made of two-dimensional representations that lack many details. In addition, Presnell [19] and Avian and colleagues [18] reported astonishing results in functional anatomy that challenged old paradigms such as the gut-sac digestive cavity in Cnidaria and Ctenophora and provided evidence for more complex digestive systems with specialized openings, albeit only appearing in much more evolved organisms [20–22]. The prevailing theories on the evolution of metazoan assume that Cnidaria and Ctenophora (which are placed at the base of the evolutionary tree of Animalia, with ctenophores even proposed as a sister group [23,24]) have simple, single-cavity digestive systems [25], while the first through-gut appeared in Bilateria [26,27].

The order Rhizostomeae certainly includes a group of taxa that are among the most representative in the coastal macro-megaloplankton of all seas, both in terms of size (up to a few meters) and occasional abundance (the so-called jellyfish blooms)

[28–30]. *Cotylorhiza tuberculata* (Macri, 1778) (Rhizostomeae, Kolpophorae, Cepheidae), usually blooming during summer [31], is a common species in the Mediterranean Sea [32–36] which hosts photosynthetic symbionts that play a fundamental role in its survival [35,37]. These jellyfish may have a potential top-down effects on planktonic communities by selectively feeding on diatoms, ciliates, larvae of some mollusks and copepods [36]. In this work, we chose *C. tuberculata* as a target species for both its abundance and distribution in the Mediterranean Sea, and its genetic distance (same order, different suborders) [38] with *Rhizostoma pulmo* (Macri, 1778) (extensively redescribed by Avian and colleagues [18]), to investigate possible similarities and differences in anatomy and function. Our aim was to (1) test whether the protocol developed and proposed by Avian and colleagues [18] for *R. pulmo* (resin endocasts followed by X-ray computed microtomography (µCT)) is applicable to other species to test its flexibility and versatility; (2) redescribe the internal and functional anatomy of this species, on the basis of the 3D resin endocasts and the microCT images, focusing on the gastrovascular system.

## Results

### Morphological analyses

Using dissections, contrast injections, resin casts, and X-ray µCT, we were able to produce an updated and very detailed three-dimensional description of the gastrovascular structure of *Cotylorhiza tuberculata* (Fig 1).

### Umbrella and stomach

The central cavity of *C. tuberculata* is usually described as circular, but its shape is indeed more complex. The upper part of the stomach, just below the thicker central part of the exumbrella, is effectively circular, and symmetric, with a convexity facing the exumbrella. The lower part, however, is provided with four distal, perradial, kidney-shaped protuberances that have lateral concavities (Fig 2A-2C). These protrusions partially extend in the arm disk, into the subgenital sinus space (Fig 2B), each one containing the more lateral parts of the four interradial folded gonads with proximal gastric cirri stripes. The floor of the stomach consists of a very thin membrane and there is no central opening. In the center of the stomach floor, the surface has a Y-shaped relief structure, with opposite bifurcations clearly visible (Fig 2B, black arrow). Further V-shaped folds are visible at their distal apex (Fig 2B, yellow arrow). The canals that communicate with the manubrium are eight, flattened and originate under the lateral extensions of the gonads (Fig 2B, red arrows; 2D).

Below the stomach is the subgenital sinus (or cavity), a single cruciform cavity, with four ovoid to heart-shaped interradial ostia (in this case the apex is exumbrellar) (Fig 2E, 2F; S1 Fig, asterisks), alternating with four perradial basal pillars which contribute to the octagonal shape of the oral disk.

The most effective way to reconstruct and to maintain the correct volumetric ratio of the stomach-subgenital sinus (Fig 2C, 2D) was to remove the umbrella above the stomach and carefully pour the resin (to avoid the pressure of the syringe) and then, once polymerization had taken place, pour a silicone resin through the ostia into the sinus. It is interesting to note that the cast of the subgenital sinus shows a greater cruciform thickness, but there are four flat and thin expansions between the cruciform arms (Fig 2D, asterisks), with eight clefts corresponding to the origins of the oral arm canals (Fig 2D, arrows).

### Umbrellar canal system

The umbrellar canal system consists of 4 perradial and 4 interradial (all rhopaliar) canals and a number of adradial canals varying between 7 and 10–11 per octant, depending on the size of the specimen (Fig 3A).

Counting the adradial canals, which originate from the stomach at the point of contact between the upper circular portion and the lower portion with the four kidney-shaped extensions, is very difficult due to the frequent dichotomies (Fig 3B) and/or ramifications. Therefore, we decided to count only the openings of the canals on the sides of the stomach (Fig 3A,

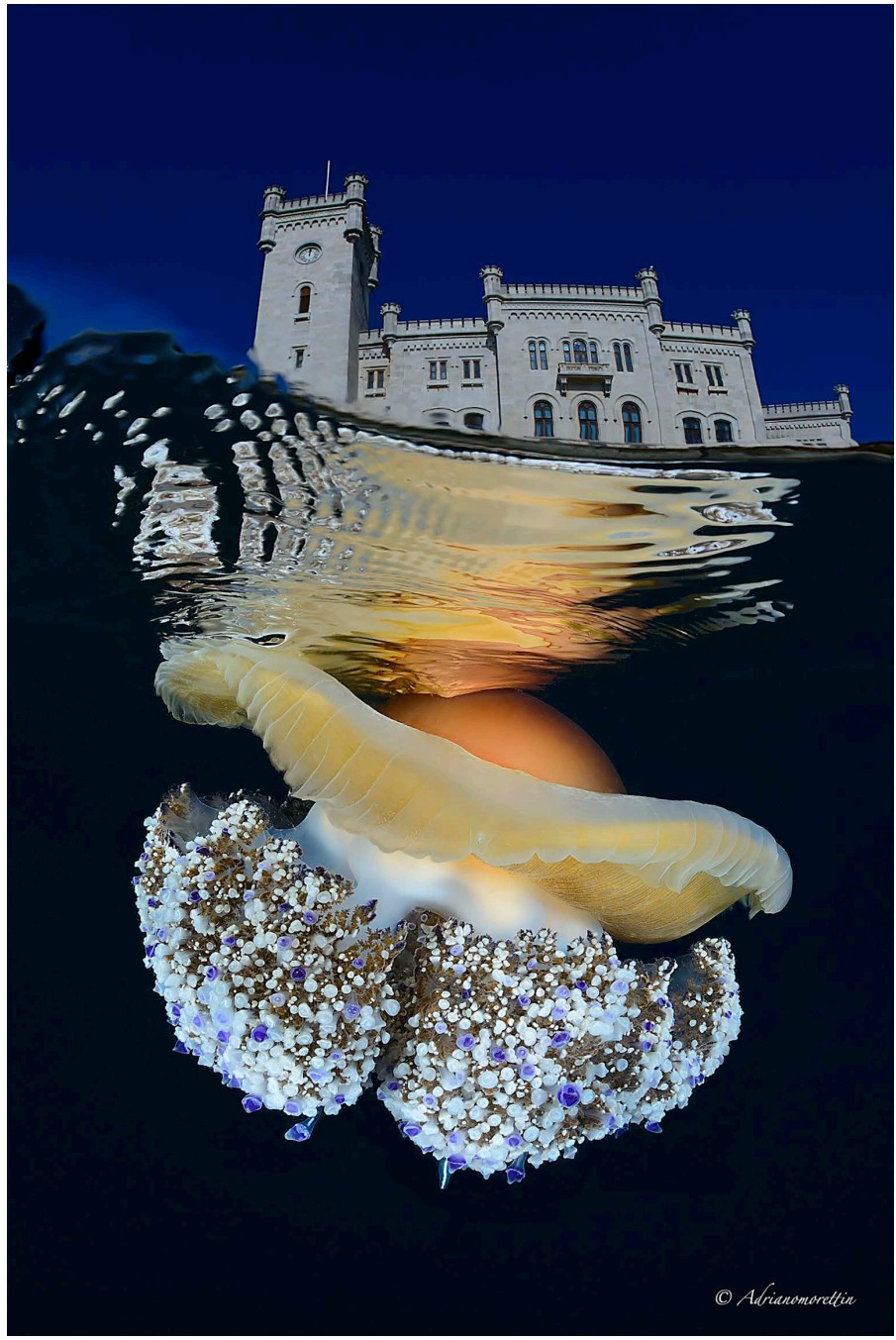

**Fig 1. *Cotylorhiza tuberculata*, adult specimen approximately 30 cm in diameter.** Photo taken in the WWF Miramare Marine Protected Area, Trieste, Italy (Miramare Castle in the background). The picture is published under a CC BY license, with permission from the author Adriano Morettin.

black arrows). All canals are strongly flattened at the oral-aboral axis, and highly anastomosed along their entire length up to the edge of the umbrella, with no evident ring canal. The numerous anastomoses begin a few millimeters from their origin, with the exception of the interradial canals, which have no connections in their proximal part (less than 1 cm in

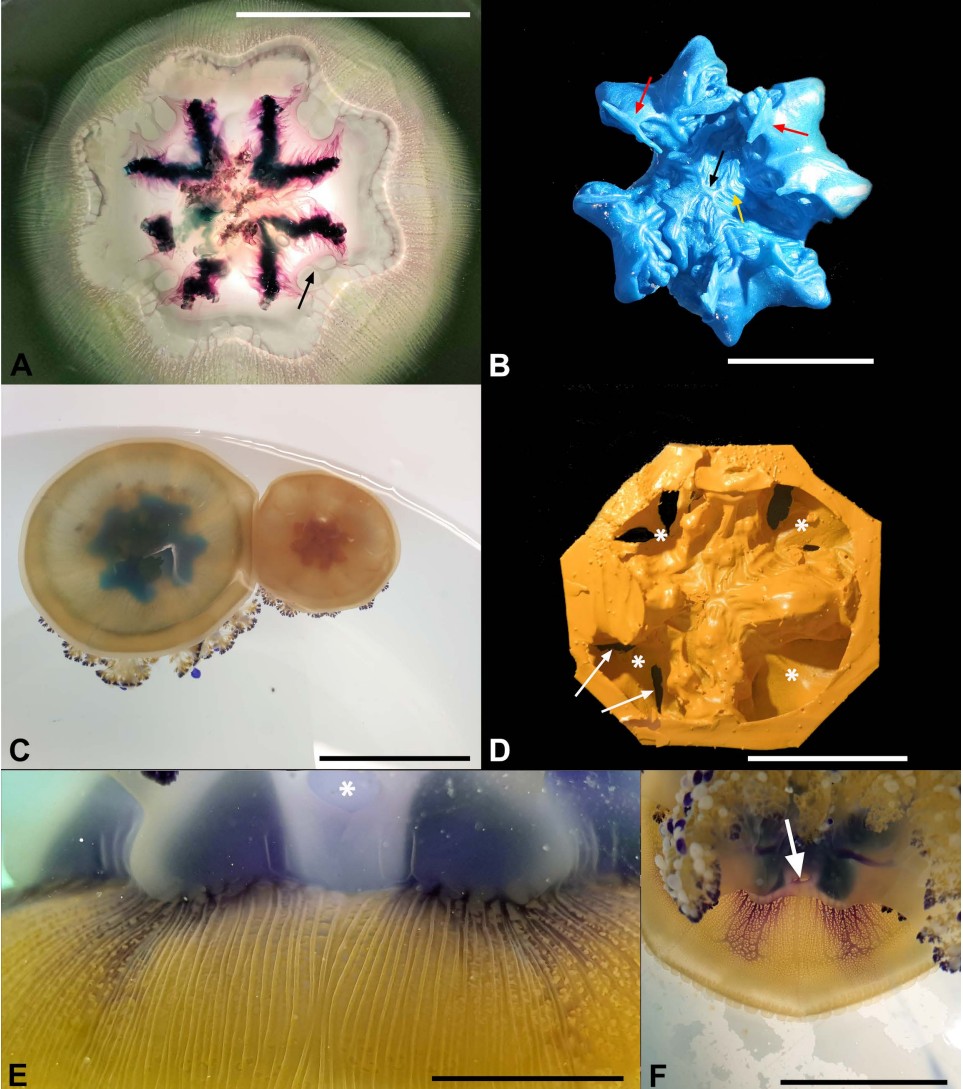

**Fig 2. Anatomy of the stomach and subgenital sinus. (A)** Transverse dissection of a stomach at the level of the four kidney-shaped protrusions, subumbrellar view. Arrow indicates one of the interradial canals. Scale bar=6 cm. **(B)** Endocast of one stomach, subumbrellar view. Red arrows indicate two of the oral arms canals. Black arrow indicates the Y-shaped relief structure present in the central area, yellow arrow indicates bifurcations. Scale bar=2.25 cm. **(C)** Two young living specimens with stained stomachs (Methylene blue and Neutral red stained, respectively). Scale bar=10 cm. **(D)** Siliconic endocast of the subgenital sinus of the same specimen of **(C)**. Exumbrellar view. Arrows indicate two of the exit points of the oral canals. Asterisks indicate the thin extensions corresponding to the stomach's protrusions in **(C)**. The octagonal edge is an artifact due to the shape of the container used. Scale bar=2.5 cm. **(E)** Adradial canals stained with toluidine blue, subumbrellar view. Interradius (and one of the subgenital ostia, asterisk) just in the middle of the picture. Scale bar=0.5 cm. **(F)** Specimen stained with toluidine blue, arrow indicates the subgenital ostium. Scale bar=4 cm.

the largest specimens). Both the perradial and interradial canals show an expansion in width after about 1 cm from the origin, which then decreases in the last distal third (Fig 3A, blue and red arrows, respectively; 3C). We have not detected the presence of a distal annular canal. However, approximately in the distal half to two-thirds of the canal system, we observed a circular, irregular, and sometimes larger, perpendicular canal (as a part of the anastomosis network) running along the entire umbrella, thus resembling an inner pseudo ring canal (Fig 3C). Some zooxanthellae clusters are detectable in correspondence with the lateral margins, or blind-ended branches, of the canals (Fig 3A, 3B).

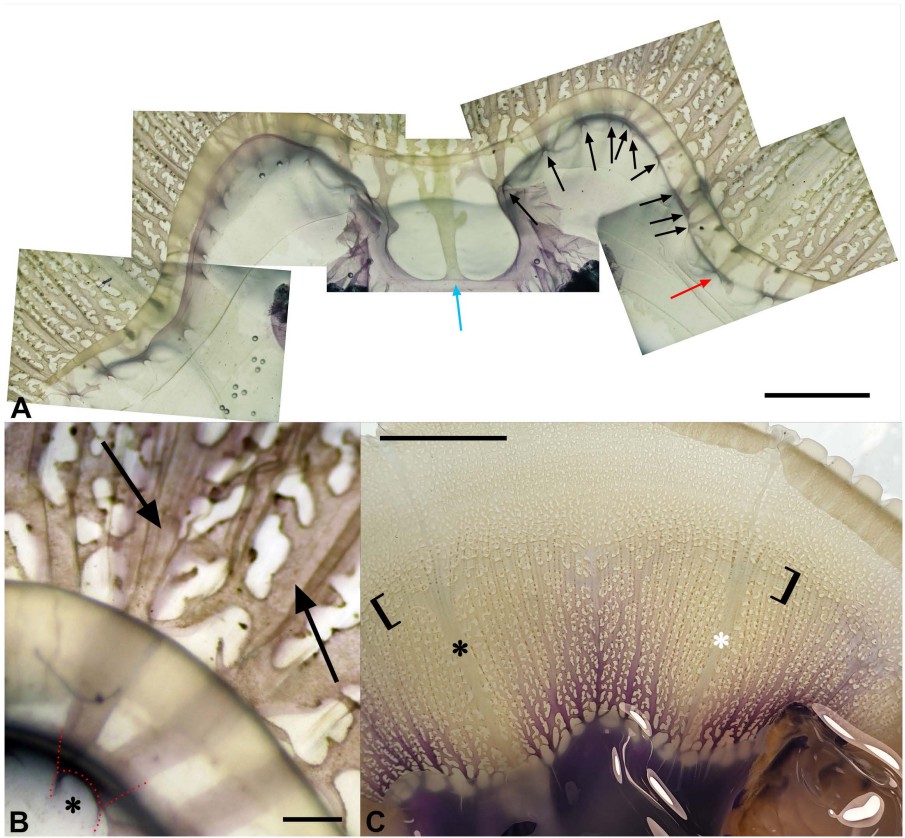

**Fig 3. Dissection of a stomach immediately below the origin of the umbrellar canals. (A)** Subumbrellar view. The red arrow indicates one of the perradial canals, the blue one the interradial canal, the black ones the adradial canals. Scale bar = 1 cm. **(B)** Detail of **(A)**, showing a pair of adradial canals (arrows) with a single origin (asterisk and red dotted line). Scale bar = 1 mm. **(C)** An octant with the adradial canals stained with toluidine blue (white asterisk = perradial canal, black asterisk = interradial canal). Scale bar = 2.5 cm. The square brackets show the widening and fusion of some canals and the corresponding canal network, which simulates a kind of "pseudo ring canal". The darker dots along canals are zooxanthellae clusters.

## Manubrium canal system

Eight canals originate from the floor of the stomach next to the most lateral portions of the gonads (Fig 2B; 4, 5A, 5B). In section they are symmetrical, flattened, with a maximum diameter of a few millimeters, with two slight terminal dilatations (Fig 5A, 5B; 6A-6C, white arrows).

Immediately after emerging downward from the subgenital sinus, they expand considerably and even reach a diameter of 1 up to 2 cm in larger specimens (Fig 4, 5A, 5B). In section, one canal is tubular distally, and very flattened in the proximal inner/lower part (Fig 6A-6C).

When analyzing the transverse sections performed in the proximal part of the oral arms, it was possible to observe a roughly median area where the two stripes of gastrodermis are in contact (Fig 6A, 6B, red arrows), but this apposition does not seem to be structurally stable, as sections with a patent lumen were also observed in this area and the two gastrodermic sides can be easily separated with a needle. However, the gastrodermis median area (the adhering stripe) has a wrinkled appearance, which could indicate adaptation in this area for increased adhesiveness (Fig 6A-6C, red arrows). The simulation of flow velocity performed on one of the oral arms virtually isolated from the skeletonization (Fig 7C), highlighted an area of high flow speed on the outer side of the oral arm canal, followed by the inner side, while the median

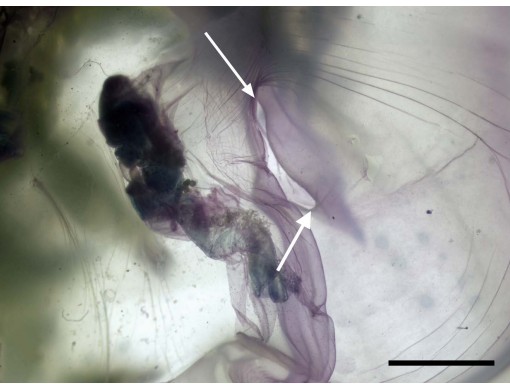

**Fig 4. Origin of the oral arm canal from the stomach.** Magnification of the stomach floor just under the lateral end of a gonad, slightly contrasted with toluidine blue, exumbrellar view. A full opening, the portion of the gonad has been previously displaced. The arrows point to the slightly wider ends. Scale bar = 2 mm.

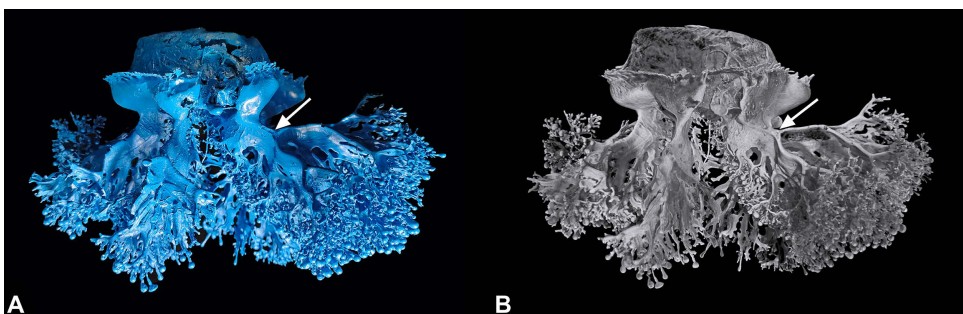

**Fig 5. *Cotylorhiza tuberculata* gastrovascular system of the manubrium. (A)** Resin endocast of the whole gastrovascular system of a specimen of 35 cm in diameter. **(B)** 3D rendering obtained by X-ray microtomographic data of the resin endocast showed in **(A)**. Tomographic reconstruction. The arrows in both **(A)** and **(B)** indicate the point of bifurcation of the two distal second-order branches.

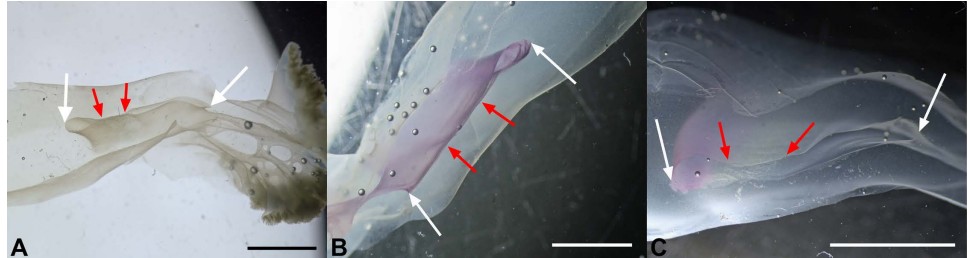

**Fig 6. Sections of the oral arms.** Sections performed in the proximal portions, before the first bifurcations. Specimen in **(A)** has a diameter of 17 cm. **(A)** Section through an oral arm canal, lateral portion on the left. White arrows indicate the full extent of a single canal; red arrows highlight an area of apposition. Scale bar = 0.5 cm. **(B)** Same type of section as in **(A)**, from a specimen measuring 20 cm in diameter, slightly contrasted with toluidine blue staining. Lateral side is at the top right. White and red arrows indicate the same features reported in **(A)**. Scale bar = 1 cm. **(C)** Section of another oral arm from the specimen shown in **(B)**, with the lateral side on the left. Arrows as in **(B)**. Scale bar = 1 cm.

stripe is clearly visible (there is a single point, probably where the two layers are not in contact, only where the flow velocity is equal to that of the lateral sides, Fig 7C, arrow).

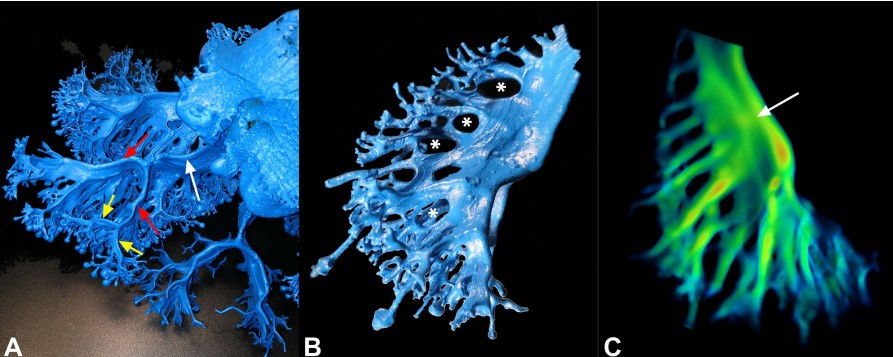

**Fig 7. Endocasts and skeletonized oral arm. (A)** Detail of the endocast of Fig 5. The arrows indicate the branches of 1st, 2nd and 3rd order (white, red, and yellow respectively). **(B)** Another oral arm endocast of a 30 cm diameter specimen. The asterisks indicate the presence of anastomoses between the lateral median branches. **(C)** A skeletonized oral arm of the specimen in **(A)**, where the flow velocity (red = faster, blue = lower) was simulated. The arrow indicates a point of a relatively higher flow point between the two lateral portions.

The eight oral arms and their associated canals run intact for about half their length in the larger jellyfish, and about 70–75% in the smaller jellyfish (15 cm in diameter), after which they are divided into a total of 16 distal portions of the oral arms. Each distal portion can lead to a second and a third order dichotomization (Fig 7A). Ramifications that extend along the entire oral arms, originate from the flattened part of the canals, may be partially anastomosed (Fig 7B) and all develop on the inner/lower side of the manubrium. All these branches may terminate in many openings or in club-shaped and sucker-shaped ends containing blind-ended canals with a white or purple/violet colored tubercle (the distal part) (Fig 1). Some of these club- or sucker-shaped ends, usually purple, can be several centimeters long (up to 5 cm in larger specimens).

This resin endocast also revealed the finest features of the manubrium and the oral arm canals (Fig 5A, 5B; 7A, 7B). In the central area of the manubrium, just below the subgenital sinus, there is a peculiar canal system connecting all eight canals of the oral arms. From a subumbrellar view, a structure can be recognized that consists of two "specular Y-shaped" canals that are connected at the base (see S2 Fig). The four lateral branches then dichotomize further to form eight canals that merge with the proximal parts of the oral arm canals. All these canals are drop-shaped in section in younger specimens (Fig 8A-8C), in older individuals they mirror the shape of the canals of the oral arms on a small scale, with a thicker exumbrellar part and a flattened subumbrellar part (Fig 8E). This structure with one to four to eight central canals was observed in all jellyfish analyzed, regardless of size and sex.

These double-Y canals have a few protrusions in adults, many of which end blindly and insert into club-shaped digitations, whereas no protrusions were found in the endocast of the younger and smaller specimen (Fig 8A-8C). These digitations vary greatly in length, from a couple of millimeters to a centimeter or more (Fig 8D-8F).

## Morphometric analysis of microtomographic data

Topological characterization by skeletonization and skeleton analysis has been carried out on the best reproduced single oral arm for two specimens (15 and 34 cm in diameter), as the smaller one (13 cm in diameter) had no oral arm completely filled with resin (Fig 9, S1, S2, S3 Video).

Table 1 summarized the most important data of the skeleton analysis (all canal terminations were considered as endpoints). It should be noted that in the adult specimen multiple openings were "connected" by artifact single droplets of resin, which were then counted as single endpoints by the software. However, in most cases, it was observed that the droplet connected only two endpoints, therefore the number of openings is slightly underestimated.

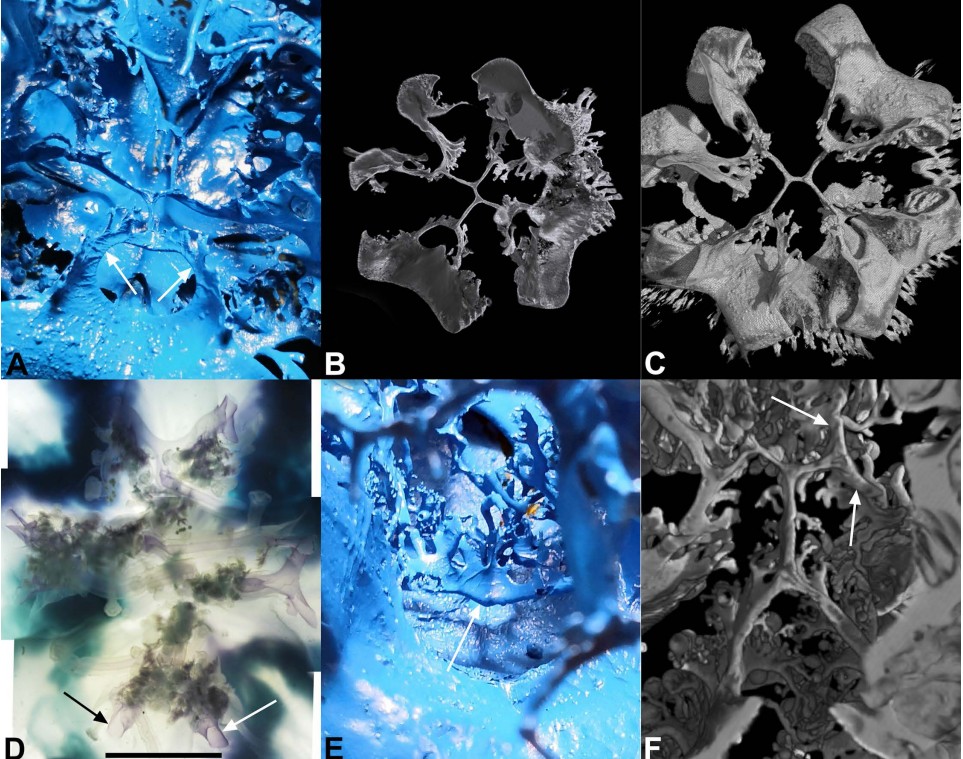

**Fig 8. Central area of the manubrium. (A)** Endocast of a specimen with a diameter of 13 cm, showing the central canal system with 1-4-8 canals, subumbrellar view. The arrows indicate two of the eight lateral branches. **(B)** 3D rendering of the endocast from **(A)**. The virtual cutting plane runs exactly through the central canal system. Exumbrellar view. **(C)** The same 3D rendering of **(B)**, but in subumbrellar view. The inner part is still free of branches. **(D)** Section of the central area of the manubrium of a specimen with a diameter of 28 cm, slightly contrasted with toluidine blue, in subumbrellar view. Several branches and club-shaped digitations are recognizable. The arrows indicate two of the eight canals. **(E)** The same central area from the 35 cm diam. endocast, in lateral view. The arrow indicates the median canal, carrying branches (upside-down view). **(F)** 3D rendering of the endocast in **(E)**, exumbrellar view. The arrows indicate two of the eight distal ramifications.

**ED = Euclidean distance**

Comparing the data between the young and adult oral arm (Fig 9E, 9F), there is a clear increase in all the data obtained, with the exception of the ED/length ratio, which remains virtually unchanged. This stable ED/length ratio in *C. tuberculata* suggests that branching geometry is determined early in development and scales proportionally with size.

As far as the number of endpoints is concerned, we estimated that a jellyfish (with 8 oral arms) has about 952 endpoints when it is young (specimen with a diameter of 15 cm) and 5632 when it is fully developed (specimen with a diameter of 35 cm), taking into account that the data were obtained by multiplying the endpoints found in a single oral arm (Table 1). These calculations may overestimate the number of actual endpoints as not all endpoints are to be considered as openings to the outside (the so-called generic "mouthlets"), as the computation also included the blind ends of the club-shaped appendages (at least one hundred per oral arm in adult jellyfish) and, in the central part, of the brood carrying filaments.

## Functional Anatomy

Experiments in which small amounts of stained solutions were injected into the stomach (S3 Fig) of young living specimens (diameter between 10 and 15 cm) have consistently shown that the outflow affects only the most lateral, slightly

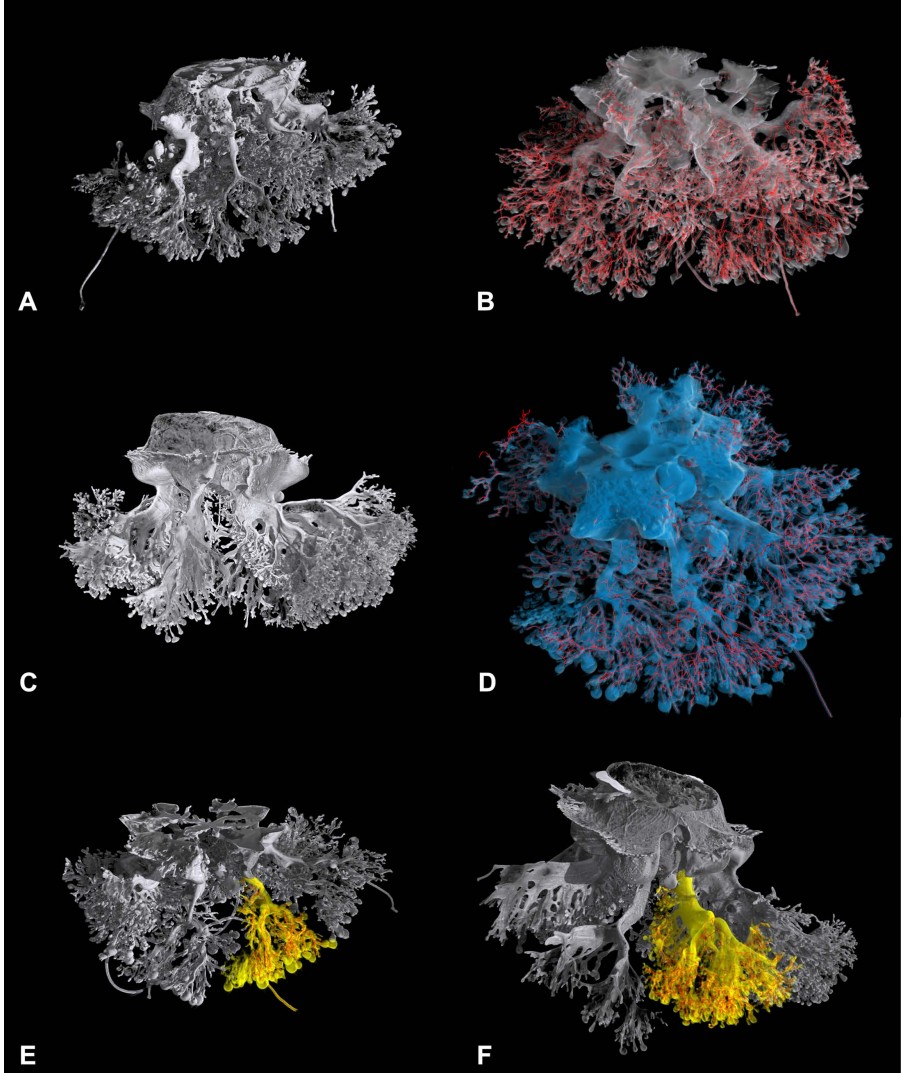

**Fig 9. Quantitative analysis on the gastrovascular system. (A)** 3D rendering of the entire cast of the gastrovascular system of the young specimen (diameter 15 cm) measured via X-ray µCT. **(B)** Isosurface rendering showing the superimposed (red) segmented gastrovascular system of the entire manubrium. **(C)** 3D rendering of the entire cast of the gastrovascular system of the fully developed specimen (diameter 35 cm) measured by X-ray µCT. **(D)** Isosurface rendering showing the overlied (red) segmented gastrovascular system of the entire manubrium (pale blue, exumbrellar view). **(E)** Volume rendering of a single oral arm (yellow) with overlaid isosurface rendering of the segmented gastrovascular system (red) in the young specimen. **(F)** As in **(E)**, in the fully developed specimen.

dilated part of the oral arm canals (Fig 10A-10C; S4-S5 Video). The inner elongated, flattened parts are not involved in the outflow (apart from some artifacts due to incorrect injection of the dye into the stomach).

This flow is relatively fast, and it usually takes 5–7 minutes to reach the openings of the 16 distal dichotomized parts of the oral arms (Fig 10A-10C). The color remains confined to the lateral part of the oral arm canals for a period of about one hour and then slowly spreads to the proximal, inner part. Although there is no physical separation between the two flows, the centrifugal flow appears to involve the more lateral, outer part of the oral arm canals and the openings in the distal, dichotomized part of the oral arm canals (Fig 11, blue arrows). Centripetal flow appears to involve the openings in the

**Table 1. Summary of the skeleton analysis of a single oral arm of a young (diameter 15 cm) and fully developed (diameter 35 cm) specimen.**

|  | 15 cm diam. | 35 cm diam. |
|---|---|---|
| *Average branch length [mm]* | 2.59 | 4.32 |
| *Average branch Euclidean Distance [mm]* | 2.06 | 3.51 |
| *ED/Length* | 0.80 | 0.81 |
| *Total number of branches* | 411 | 1442 |
| *Total number of junctions* | 224 | 688 |
| *Total number of endpoints* | 119 | 704 |

ED = Euclidean distance.

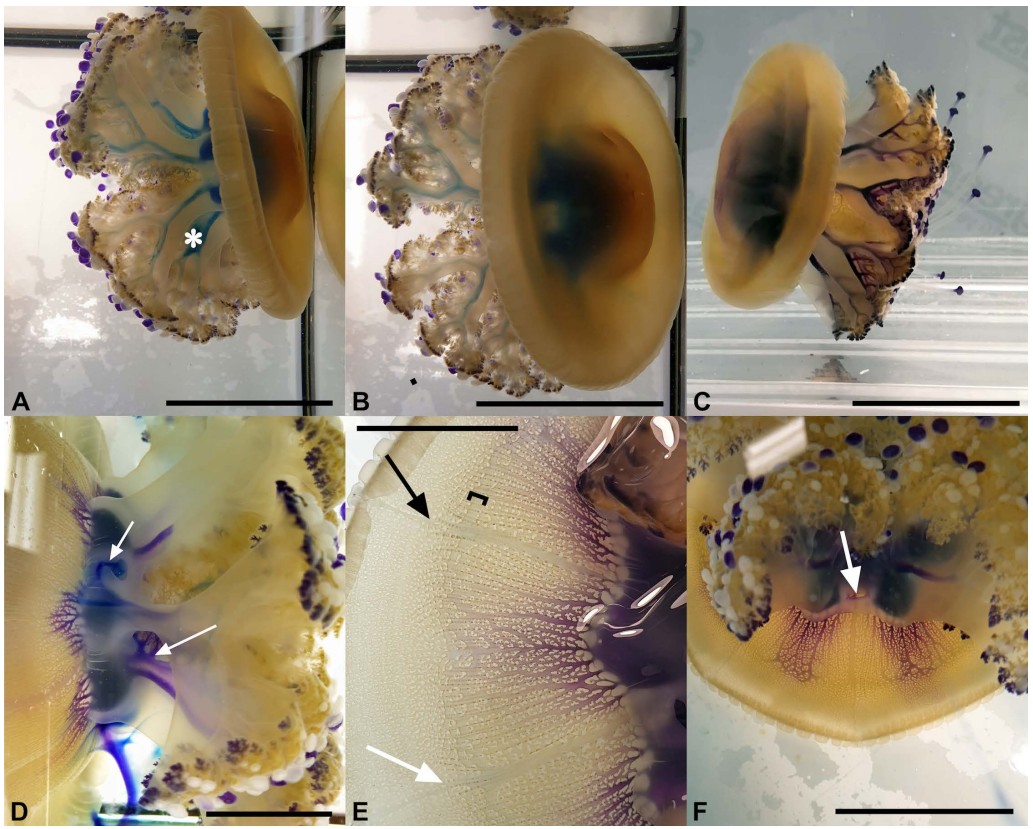

**Fig 10. Functional anatomy of the gastrovascular system. (A, B)** Experiment from July 2022. Specimen injected with methylene blue. Visible lateral parts of the stained oral arm canals. The asterisk indicates an artifact due to the inclination of the needle during injection. Scale bars = 6 cm. **(C)** Experiment performed from August 2023, carried out with toluidine blue, at about 15 minutes after the injection. The lateral parts of the oral arm canals and the branches present after the dichotomy of the oral arms are recognizable. Scale bar = 8 cm. **(D)** Another experiment (August 2023), toluidine blue, after 5 min. The arrows indicate the staining of one of the inner part of an oral arm canal (artifact) and the release of dye from the subgenital cavity (perforation of the stomach floor). Scale bar = 4 cm. **(E)** July 2022, diffusion of toluidine blue at about 10 minutes through the adradial canals in an octant. the arrows indicate the perradial (black) and interradial (white) canals. The square bracket shows the widening of some canals and the corresponding canal network, which simulates a kind of "pseudo annular canal". Scale bar = 2.5 cm. **(F)** Another experiment like **(E)**, but after 30 min. The arrow indicates one interradial subgenital ostium. Scale bar = 4 cm.

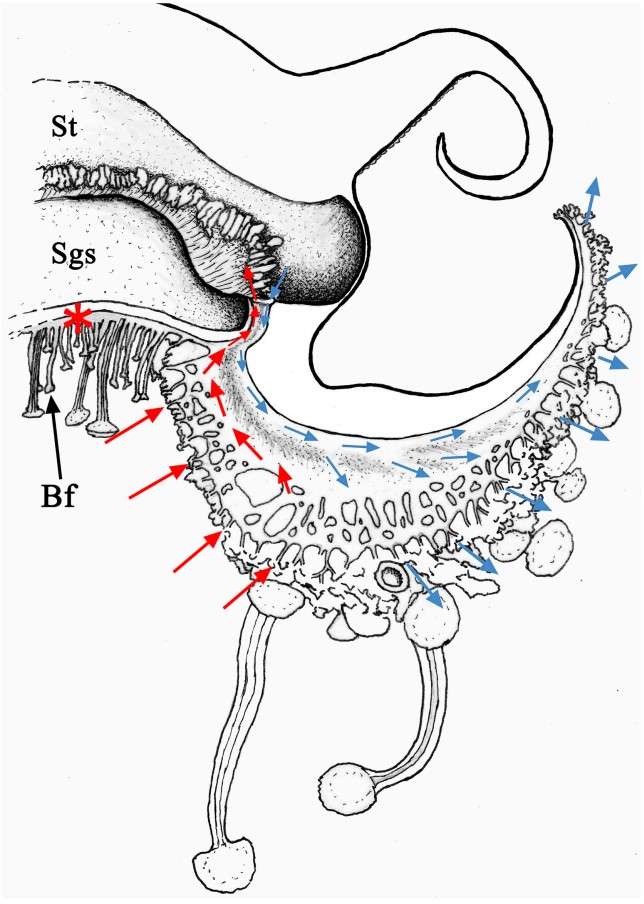

**Fig 11. *Cotylorhiza tuberculata*, schematic representation of the gastrovascular circulation.** Drawing based on adradial sections. The oral arm is represented without the proximal and distal bifurcations, and with a simplified system of branches and openings. From top to bottom: the stomach (St) with on the bottom right the origin of the oral arm canal; the subgenital sinus (Sgs); the peculiar central canal system connecting all eight canals of the oral arms (red asterisk) and carrying the brood-carrying filaments (Bf) and some oral arm appendages. The shaded area in the oral arm indicates the apposition surface between the two oral arm canal distal parts. Red arrows, incoming flux in the internal/lower part of the oral arm canal, blue arrows outgoing flux in the external/upper part of the oral arm canal.

more proximal part, the one preceding the dichotomy of the oral arm canals, and the more internal, flattened parts of the oral arm canals (Fig 11, red arrows).

The second experiment, in which one specimen was immersed in a small tank (Fig 12A) of colored seawater, showed, after leaving the jellyfish for about ten minutes, the presence of an influx involving the innermost/lower portions of the oral arm ramifications and the relative flat parts of the oral canals, and that the dye had reached the stomach. Recording the experiments was extremely difficult, mainly due to the time wasted (the stain intensity tends to reduce quickly), the stress induced by the rinsing passes, and the masking effect due to the residual dye on the jellyfish's epidermis.

In the umbrella, the stain diffuses centrifugally through the adradial canals (Fig 10E, 10F). In this area, however, the flow rate is rather slow. A first series of experiments carried out in July 2022 with specimens collected off the coast of Piran (Slovenia) transported to the laboratory in Trieste (Italy) and acclimatized there for 2–3 days without food, had shown a very slow flow. Half an hour after injection, the adradial canals and the associated anastomoses were stained only in the proximal half and barely reached the inner "pseudo ring canal". The pseudo annular (= ring) canal (Fig 10E)

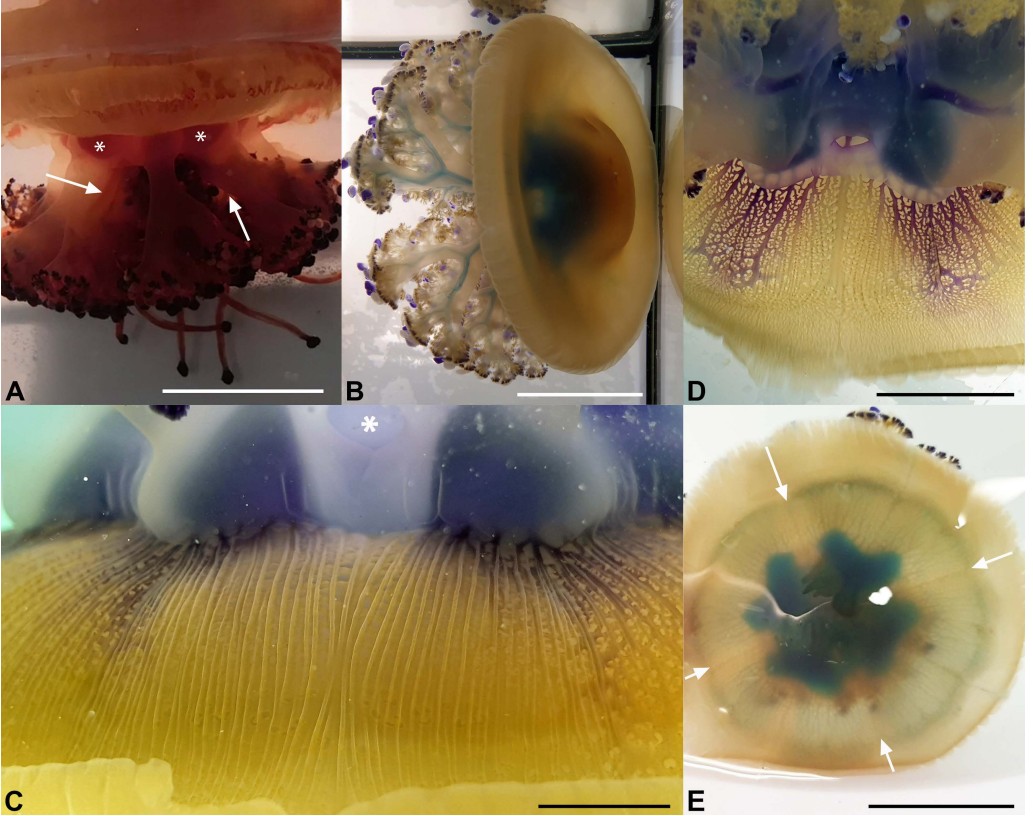

**Fig 12. Functional anatomy of the gastrovascular system. (A)** Experiment carried out by immersing a specimen in an aquarium with stained sea-water (Neutral red). The arrows indicate the inner part of two of the oral arm canals. The asterisks indicate the stain in the stomach. Scale bar = 5 cm. **(B)** Experiment from August 2023, methylene blue, as in Fig 10A, 10B. Scale bar = 4 cm. **(C)** Onset of diffusion of toluidine staining through the adradial canals after about 7-8 minutes, subumbrellar view. Interradius (and one of the subgenital ostia, asterisk) exactly in the middle of the picture. Scale bar = 0.5 cm. **(D)** Stain extension after 10 minutes from the injection, subumbrellar view. **(E)** Exumbrellar view of a specimen injected with methylene blue, after 30 and more minutes. The main diffusion of staining is also visible in the interradial canals, but not yet in the perradial ones (arrows). Scale bar = 4 cm.

corresponds roughly to the limit of the rigid part of the umbrella, just before the bending point. It is noteworthy that circulation in the most distal part of the umbrella canal system is much slower and a complete diffusion of the dye to the whole umbrella margin was never observed (in a few cases, only some octants per jellyfish were totally filled). The same series of experiments was repeated the following year (August 2023), but this time directly in the Piran Aquarium, with specimens caught the day before. This time the dye flowed a little faster, and about half an hour after injection it reached the inner "pseudo ring canal". However, in all the experiments that we carried out (2022 and 2023), the diffusion of the dye in the anastomoses initially only affected the adradial area, then it spread slowly and scarcely in the more proximal portion of the perradial canals, but not in the interradial canals. Within half an hour, the stain reached the inner "pseudo ring canal" (Fig 10F, 12D, 12E), and spread towards the inter- and perradial canals (Fig 11, S3, black and white arrow, respectively). In rare cases it was possible to see the staining of the distal network canals up to the edge of the umbrella. Normally, about 1 h or more after injection, once the staining reached the per- and interradial canals, it began to drain into the stomach and the adradial anastomoses became clearer again (if methylene blue was used as a dye) (Fig S3). The flow rate was rather slow, and the exchange of the volume of circulating liquid in the network of umbrella canals required longer times, especially in the distal anastomoses.

## Discussion

Anatomical studies of jellyfish have been a time-honored but niche discipline, often constrained by significant limitations [18]. Despite general scientific advances in the 21st century, many of the techniques and descriptions used in this field date back to the 19th century and are still considered undisputedly valid.

The innovative protocol by Avian and colleagues [18], which yielded remarkable results for *Rhizostoma pulmo*, raised open questions about its potential reproducibility and applicability in other species. The success of this technique in *Cotylorhiza tuberculata* shows that it is indeed versatile and can be easily applied or adapted to other rhizostomean jellyfish, not only to young and relatively small specimens, but also to adults (diameter of 35 cm, present study).

*Cotylorhiza tuberculata* is a rhizostomean (Suborder Kolpophorae, family Cepheidae) jellyfish that was first described without a specific name by Macri [39] in his work on the "Polmone di mare" (now *Rhizostoma pulmo*; Macri, 1778, p. 20–22), later named and described as "*MEDUSA tuberculata*" in the XIII ed. of Systema Naturae edited by Gmelin [40]. *C. tuberculata* is relatively common in coastal Mediterranean waters and Northwest coast of Africa [41], and various studies on its biology, anatomy and life cycle are available [28,35,37,41–53]. In particular, the study by Kikinger [35] contains a description (and a fine drawing that has become a classic) of the internal anatomy of *C. tuberculata*, which we have used as a comparative term for our analyses.

### Gastrovascular system anatomy

We described a more complex shape of the stomach than a simple generic rounded cavity, being the lower portion characterized by four perradial kidney-shaped extensions (Fig 2A, 2B), which occupy part of the space of the subgenital sinus (Fig 2B) and contain the more lateral parts of the four interradial folded gonads with proximal gastric cirri stripes. The inner space of the stomach and the underlying subgenital cavity are separated by a thin and delicate membrane that constitutes the floor of the stomach, on which the four gonads develop. Furthermore, we were able to show the presence of thin extensions under the four kidney-shaped expansions of the stomach, which call into question the "traditional" [35] cruciform description of the subgenital sinus, now observed as more circular. This suggests that the subgenital cavity is reminiscent of the theoretical circular shape, with its four cruciform pillars alternating with the four kidney-shaped protrusions of the stomach resting on the floor of the subgenital sinus. Since the resin injection experiments performed in the subgenital sinus caused a dilation in these four areas, we suggest that the cruciform shape in living specimens is only functional and is determined by the normal physiological shape of the stomach when it contains circulating seawater (so that the four perradial areas are kept flattened by the stomach). Another intriguing issue concerns the impression or scar that can be seen in the central area of the floor (the subumbrellar membrane) of the stomach (Fig 2B), with two opposing Ys and eight subsequent bifurcations. This scar recalls the early stages of formation of the subgenital sinus, as already observed in ephyrae – metaephyrae up to 1 cm in diameter, where four interradial invaginations develop until they fuse centrally and close the central mouth [49]. In metaephyrae of 10 mm in diameter there are four large oral arm canals, the oral arms are four in the most proximal portion, but distally they are already eight (with eight oral arm canals; their separation starts from the most distal margin). Although it was not possible to observe the subsequent stages of development, it seems conceivable that the dichotomization of the four initial oral arms proceeds until reaching the basal pillars, which are four even in adult specimens. However, the dichotomization of the oral arm canals proceeds within the basal pillars until reaching the floor of the stomach, bringing the adradial oral arm canals to eight (Fig 2B, S2 Fig). This pattern of scars is perfectly specular to the arrangement of the central canals of the oral arms (Fig 8) that we have described. This series of canals has already been described by both Kikinger [35] and Gibbons and colleagues [41], but in a summarizing manner and with little detail. An initially quadripartite arrangement is evident, and, based on these characteristics, the planes of symmetry in this species are only two, orthogonal, and not radial. However, scar figures are not uncommon in other rhizostomeans. *Rhizostoma octopus* [54,55], for example, present a similar structure in the central subumbrellar portion of the oral arms (although, in this case, the suture is central, but below the residual proximal portion of the cruciform canal of the manubrium).

In the umbrella, we recorded a different pattern of the canal system compared to the more uniform canal topography described in Gibbons and colleagues [41], where the canals are well differentiated proximally. In our specimens, we observed numerous anastomoses between canals, branching immediately after a few millimeters from their origin at the lateral edges of the stomach. The only exceptions were the interradial canals, in which the first part (one centimeter or less) from the origin was free of ramifications (Fig 3). According to the current state of the art, the umbrellar canals pattern of *C. tuberculata* is described as having no annular (or ring) canals. However, we report for the first time an area of increased flow that appears to functionally mimic an inner ring canal (Fig 10E, 12E; S3 Fig).

In taxonomy, the number of adradial canals is usually regarded as a common diagnostic feature, often used in identification guides, and it can be easily examined at low magnification by transparency in the order Rhizostomeae. However, the resulting number in our specimens (which is difficult to count given the high degree of anastomoses present) is variable, ranging from a minimum of 7 (in younger specimens) to a maximum of 9–11, with different values even in different octants of the same specimen. It should also be noted that observations by transparency may lead to an overestimation of the number of canals, since in some cases two adjacent canals share a single opening, as confirmed by our dissections (Fig. 3b). In such cases, we counted them as a single canal. Therefore, we suggest that the number of adradial canals counted by external macroscopic observation (without sectioning the base of the stomach) is not a strong diagnostic character, as the differences within the genus *Cotylorhiza* are really minimal (*C. erythraea* Stiasny, 1920 is described to possess 4–8 adradial canals per octant [47,54–56], while *C. ambulacrata* Haeckel, 1880 features 11–13 adradial canals per octant [41,54,55]). The canals of the oral arms of *C. tuberculata* are rather flattened in their medial part, where the gastrodermis of both sides joins, but in a rather labile manner (Fig 6A, 6B). These strips look more wrinkled than the rest of the gastrodermis, suggesting an adaptation to at least partially separate the flows between the inner and outer parts of the canals (Fig 6A, 6C), which is also evident from the flow simulation (Fig 7C). In addition, a smaller number of anastomoses (Fig 7A, 7B) can be observed in the inner part of these canals compared to the canals depicted by Kikinger [35] or Gibbons and colleagues [41].

The use of resin endocast and subsequent 3D imaging obtained from X-ray microtomographic data, allowed us to detect a series of parameters on the isolated oral arms (Table 1). In the two specimens we analyzed (15 and 35 cm in diameter), the ED/length ratio remained constant. The similar values observed within *C. tuberculata* life stages and *Rhizostoma pulmo* (ED/length ratio of 0.82, Avian et al., 2022) may indicate a convergently evolved anatomical. The number of endpoints increased with size. This calculation yielded 952 and 5632 endpoints, respectively, of which approximately 900 for the small individual and 5200–5400 for the adult specimen can be assumed to be true openings. The only possible comparison with the available literature data is with *R. pulmo*, for which Avian and colleagues [18] measured 4001 openings (oral arms plus scapulae) in a specimen with a diameter of 14.6 cm. It should be considered that the examined specimen of *R. pulmo* specimen was not a fully developed adult, as this species can normally reach a diameter of 40 cm. It is therefore evident that the two taxa differ significantly in the total number of openings. This diversity may be interpreted by taking into account their different feeding strategies. Both species are planktophagous, but, since *C. tuberculata* hosts symbiotic zooxanthellae that contribute substantially to its metabolic demands [35,37], it can be hypothesized that this species relies less on predation, which may explain the reduced number of oral openings compared to *R. pulmo*.

### Functional anatomy

In the umbrella, experiments have shown that centrifugal flows involve only the adradial canals (Fig 10D, 10E, 12D) and, once reached the pseudo-ring canal, the flow then spreads along this ring, and centripetally flows reaching first the perradial canals (Fig 12F) and, successively, the interradial ones (S3 Fig). Then, the dye centripetally diffused through these canals towards the stomach (S3 Fig), as commonly observed in other scyphozoan jellyfish [54,55,57]. Flow speeds in the umbrella are slower in *Cotylorhiza tuberculata* (over 1 hour to partially reach the umbrellar margins), when compared to the previous research on *Rhizostoma pulmo* (about 20–30 minutes to completely reach the umbrellar margins [18]).

A possible explanation could be linked to the fact that *C. tuberculata* hosts gastrodermic (=endodermic) zooxanthellae [35,37], which are distributed into the network of umbrella canals, just above the radial muscle fibers (Fig 12C). If the metabolic requirements for the pulsations are also supported by symbionts, the diffusion of liquids from the stomach up to the margins could become less important, and therefore slower.

Experiments performed by injecting dyes into the stomach of living specimens (diameters between 10–15 cm) confirmed what had already been observed in *R. pulmo*, namely that the outflow occurs in the outermost part (the most dilated) of the oral arm canals (Fig 10–12). Rarely, in the oral arm canals, the dye slowly spread from the outer to the inner canal, indicating that the apposition area between these structures is not perfectly sealed (Fig 7C), as observed in *R. pulmo* [18], which features true hemi-canals. The immersion experiment seems to confirm the observations cited above. In this case, the colored portions of the canals are the inner ones, as is the stomach (Fig 12A), supporting the theory of a bidirectional flow (centripetal in the innermost portion, centrifugal in the outer one) induced by the beating of the gastrodermal ciliated cells in the canals of the oral arms. Therefore, also in this case, the openings on the lower surface of the oral arms are to be considered as mouthlets in the most proximal area of the oral arms before the dichotomy, while those present in the distal portions perform the function of anuses. This circulation pattern corresponds to that of *R. pulmo* [18], but the centrifugal flow of the manubrium in *C. tuberculata* is faster. Although these species are notably different from a morpho-anatomical point of view (in *C. tuberculata* there is no distinct longitudinal subdivision of the oral arm canals and no separation of the perradial canals in the stomach) [18], the functional pattern is quite similar.

The family Cepheidae (Infraorder Actinomyaria) currently includes five genera: *Cephea*, *Cotylorhiza*, *Marivagia*, *Netrostoma* and *Polyrhiza*. The presence of eight oral arm canals at the level of the arm disc and the regression of the central mouth (completely absent in these species, no data are available for *Marivagia*; [58–60]) can be considered as one of the diagnostic characters of this infraorder [54,55]. In the other rhizostomeans, the oral arm canals generally originate from the remnants of the central mouth. The most recent phylogenetic reconstructions [38] indicate the Rhizostomatidae and the Stomolophidae (both Daktyliophorae) as the most phylogenetically distant taxa from the Kolpophorae (including Cepheidae). Thus, given that this circulation pattern is shared by two distantly related taxa, it is conceivable that this represents a symplesiomorphic characteristic that is more widespread within the order Rhizostomeae than previously thought. This hypothesis seems to be supported from a morphological point of view by Stiasny's [47] analysis of the oral arm canal systems of various species of Rhizostomeae, where further examples of oral arm canals with internal constriction are reported, such as *Lobonemoides gracilis* Light, 1914 (previously known as *L. robustus*) [47]. This additional observation of separate circulation and openings with specialized different functions (only inflow-only outflow) in Cnidaria provides further evidence for a through-gut like system, which also revives the debate on the digestive system evolutionary theories [18,19].

In conclusion, we have confirmed the validity of the protocol proposed by Avian and colleagues [18], with which excellent results were also obtained in *C. tuberculata*. The ease of use, flexibility and universality of this protocol could mean that it can be applied by jellyfish researchers all over the world and is not limited to selective studies.

From an evolutionary perspective, the discovery of the double canal and inner circulation provides new insights into the development of the gastrovascular system in Animalia [19,26,27,61]. Our findings may contribute to refining current views of jellyfish, which are often anatomically regarded as 'simple' compared to other Macro and Megaplanktonic organisms. From a wider point of view, they could also suggest that evolutionary trajectories are not always linear progressions from simple to complex forms, but may involve unique or rare deviations from this pattern [18,62–64].

The authors conclude with the hope that this study along with the one by Avian and colleagues [18] could serve as basis for a new worldwide wave of descriptions and updates of the functional anatomy of jellyfish, the significance of which goes beyond Cnidarian research and the implications of which have a major impact on broader scientific debates.

## Materials and methods

### Sampling area

*Cotylorhiza tuberculata* individuals (67 total specimens) were collected in two locations in the Gulf of Trieste (North Adriatic Sea): the LTER_EU_IT_056 site in 2022 (Italy), and coastal water off Piran (Slovenia) in 2022 and 2023 (Fig 13). No permits to access sampling sites were required because the areas are part of the public maritime domain. Dates of samplings are reported in Table 2.

### Sampling design

Juveniles of *C. tuberculata* usually appear in the Gulf of Trieste in early summer and adults disappear when temperatures drop during late autumn. We collected specimens of different sizes and developmental stages by diving sessions using

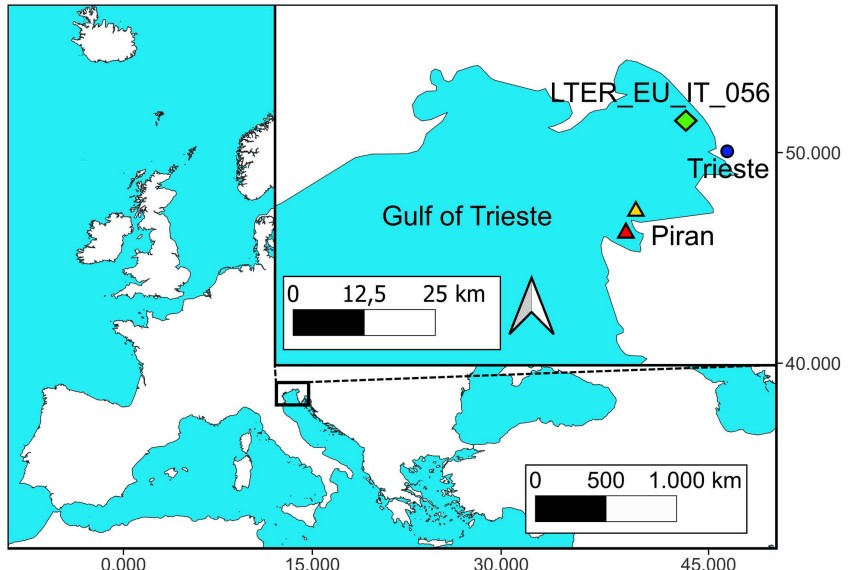

**Fig 13. Sampling map.** The top right square zooms on the sampling area in the Gulf of Trieste (North Adriatic Sea). The green diamond indicates the Italian sampling site LTER_EU_IT_056. Triangles represent the sampling sites at Piran, with the yellow triangle referring to the coastal waters sampling site and the red triangle to the shore sampling site. The blue circle marks the location of Trieste (It), included as a geographical reference point. Map source Natural Earth (Free vector and raster map data @ naturalearthdata.com), which material are in the public domain, no permission to use is needed.

**Table 2. List of collected samples and locations for *C. tuberculata*.**

| Locality | Time | Specimens n (Diameter) |
|---|---|---|
| LTER_EU_IT_056 (It) coastal waters. 45.70' N, 13.71' E | July 12, 2022 | 2 adults (> 25 cm) |
| Piran (Slo) coastal waters. 45°31'54.02"N 13°33'39.33"E | July 13, 2022 | 37 juveniles (5–20 cm) |
| Piran (Slo) shore. 45°31'39.57"N 13°33'59.31"E | September 9, 2022 | 2 adults (25 cm) |
| Piran (Slo) coastal waters. 45°31'33.19"N 13°34'53.30"E | August 9, 2022 | 2 adults (> 25 cm) |
| Piran (Slo) shore. 45°31'39.99"N 13°33'58.01"E | August 17–24, 2023 | 24 juveniles (5–20 cm) |

containers to avoid physical damage to jellyfish. A total of 61 juveniles (diameter up to 20 cm) and 6 adult jellyfish (diameter over 25 cm) were sampled (Table 2). All research present in the manuscript (being on jellyfish) did not require ethical approval. In 2022, half of the specimens were reared in a single tank (Aurelia-80, Reef-Eden), the other half were fixed in 5% formalin-saltwater solution and then stored. In 2023, all specimens were kept in buckets and processed in the days immediately following sampling.

## Morphological analyses

### Characterization of the gastrovascular system and functional anatomy

To describe *Cotylorhiza tuberculata* gastrovascular system we followed the protocol proposed by Avian and colleagues [18].

Methylene blue (0.5%), Toluidine blue (0.5%) and Neutral red (1%) were injected in-vivo from the apex of the umbrella into the stomach with an insulin syringe (S4 Fig). Depending on the size of the specimen, we injected approximately 1–5 ml in smaller individuals and 5–10 ml in adults, stopping when the stomach appeared full.

It is important to underline that only a small amount of stain solution should be injected to avoid excessive pressure in the stomach, which would affect the subsequent observations. The injection should also be as central and vertical as possible. If the needle of the syringe is tilted, the dye would be pressed into the corresponding side with strong pressure and thus mechanically "squeezed" directly into both oral arm canals (Fig 10A, 10D). Conversely, a low volume of dye and a weak pressure allow the dye to diffuse well through the gastrovascular circulation currents.

The internal flux circulation experiments were performed in July 14–15th, 2022, and August 17–25th, 2023, and video recorded (Canon G16 camera). When the canals were filled, the specimens were immediately fixed in 5% formalin. The oral arms were then excised and observed under a Leica Microsystems M205C Stereomicroscope equipped with a Canon G16 camera.

In a second experiment, the intake flow of *C. tuberculata* was investigated. For this purpose, for each trial, a single specimen was immersed for 10 min in a 5 L tank containing seawater dyed with 1.5% Neutral red. The specimen was then transferred in a series of three tanks filled with seawater to "clean" the outer surface and then filmed as above.

### Internal cast of the gastrovascular system

We applied the protocol by Avian and colleagues [18] to make multiple casts of the whole and single parts of the gastrovascular system to obtain the clearest visualization of the ramifications of the canals. Individuals fixed with 5% formalin were used. We chose Liquidissima Resin Pro© Epoxy resin with a 5% acetone dilution to obtain the best balance between viscosity and fluidity. Fixed jellyfish were kept in water to sustain bodyweight and avoid tissues breakdown, and the resin was injected into the stomach using a 20-ml syringe for smaller specimens, and a 50-ml syringe for larger specimens. Jellyfish tissues were gently squeezed to distribute the resin throughout the gastrovascular system. The resin hardened within 24 hours, then jellyfish tissues were digested for 24 hours at 40°C in a 20% KOH solution (sample/solution volume ratio 1:5).

Several trials have been made to reproduce the three-dimensional shape of both the subgenital sinus and the stomach and their volumetric relationships. We tried to inject resin with a syringe both into the stomach and through the ostia into the subgenital sinus. Many attempts were unreliable because when the resin was first injected into the stomach, it swelled and compressed the subgenital sinus (the membrane separating the stomach from the sinus is extremely thin and plastic). Similarly, when the resin was injected into the subgenital sinus, we obtained an inflated subgenital sinus, thick, circular and not cruciform and the stomach was squeezed.

## X-ray computed microtomography measurements

Three endocasts (two juveniles of 13 and 15 cm and one adult of 35 cm of diameter) were analyzed trough the microfocus X-ray computed tomography (µCT) technique by using the FAITH instrument of the Elettra synchrotron light facility in Basovizza (Trieste, Italy). This fully customized cone-beam CT system is equipped with a sealed microfocus X-ray source (Hamamatsu L12161, Japan) operating in a 40–150 kV energy range at a maximum current of 500 µA. For this study, a flat panel detector (Hamamatsu C11701DK) with a resolution of 2192 × 1776 pixels and a pixel size of 120 × 120 µm² was used. By adjusting the source-object-detector distance it is possible to obtain a spatial resolution close to the minimum focal spot size of the source (5 µm) for samples of different dimensions varying from 1–2 mm up to 30 cm in lateral size. For these acquisitions, we set the detector position in an offset configuration, which granted an approximately 60% increase in the lateral Field of View (~1.6 times the standard Field of View). Two set of scans were performed at both low (voxel size 84 µm) and medium (voxel size 36 µm) resolution.

The low-resolution X-ray µCT scans on the whole sample were acquired using the following experimental conditions (High Voltage setup):

- Source: tube voltage = 100 kVp, tube current = 270 µA, focal spot size = 20 µm, 0.2 mm Cu filter

- Geometry: source-sample distance = 420 mm, source-detector = 600 mm, equivalent voxel size 84 µm

- Offset detector geometry [65] with a displacement of 30%

- Number of projections = 3600, angular step = 0.1 degrees, exposure time = 0.2 s, total scan duration = 12 min.

The medium resolution X-ray µCT scans on the whole sample were acquired using the following experimental conditions (High Voltage setup):

- Source: tube voltage = 100 kVp tube current = 270 µA, focal spot size = 20 µm, 0.2 mm Cu filter

- Geometry: source-sample distance = 180 mm, source-detector = 600 mm, equivalent voxel size 36 µm

- Offset detector geometry [65] with a displacement of 30%

- Number of projections = 3600, angular step = 0.1 degrees, exposure time = 0.2 s, total scan duration = 12 min.

CERA software v. 7.0.0 (Siemens Healthineers AG, Germany) was used to reconstruct a set of 2D axial slices from the projection images (radiographs) obtained by the flat panel detector during a full 360-degree rotation of the specimen. Fiji [66] and VGStudio MAX 2.0 (Volume Graphics, Germany) were used to investigate the structure and anatomy of the three samples.

## Morphometric analysis of the X-ray microtomographic data

To obtain a comprehensive description of the gastrovascular system, we analyzed specific regions of the specimen, with different foci: 1) the entire cast, 2) the entire manubrium to compare the structure of the oral arms, and 3) a single oral arm to characterize the topology of its structures (the wings) supporting the openings of the canals.

A fully quantitative analysis is essential to accurately assess differences between jellyfish samples imaged at different developmental stages. As shown by Avian and colleagues [18], the most informative parameters are derived from the skeletonization process of the oral arms. The topological characterization via skeletonization and skeleton analysis has been carried out on the most complete and accurate cast and skeletonized oral arms to avoid possible underestimations due to incomplete injection of the resin, cast fractures and, at the same time, to avoid under- or overestimation due to the skeletonization process. This approach was preferred to a complete analysis of the entire endocast as a few oral arms were not fully recovered or filled with the resin.

 

The analysis was performed by calculating "skeleton" for each oral arm. This skeleton in this case is defined as the medial axis of the structure (i.e., the medial axis of the gastrovascular canal structure in the oral arm) calculated using the thinning algorithm [67].

The detailed analysis of the oral arms proceeded as follows: (i) selection and isolation of the most complete oral arm from each dataset, (ii) binarization using automated thresholding [68] to prepare for skeletonization, (iii) 3D Gaussian smoothing to maintain consistent medial axis positioning, followed by segmentation again, to create a noise-reduced binary volume, avoiding the need for aggressive pruning after skeletonization, (iv) 3D hole-filling to eliminate gas bubbles in the resin, and a connected components calculation to remove "floating" objects that could interfere with skeletonization, (v) skeletonization via the thinning algorithm with two cycles of pruning for the shortest branches (3 voxels), and (vi) skeleton analysis.

All steps were executed using ImageJ/Fiji software [66], with volume renderings created using VGStudio MAX 2.2 (Volume Graphics, Germany).

The topological characterization of the systems features two main concepts: the local thickness (LT) analysis, and the "skeleton" (i.e., the medial axes of the structure, in this specific case) analysis. The LT of a segmented class in a tomographic dataset is described as the calculation for each voxel (vx) of the radius of the maximum inscribed sphere in the class that contains that vx [69]. The skeleton has been measured via the "thinning" algorithm [67], as we were interested in the calculation of the medial axis of the structures. The combination of the two analyses can provide a volume where the medial axes are labeled, vx by vx, with the LT values, to provide the local quantification of the thicknesses. A similar approach has been successfully applied to characterize the pore space topology in porous materials [70].

The analysis on both the manubrium and the single arm was carried out as follows: first, the grayscale 8-bit dataset was segmented using the Otsu thresholding algorithm [71] applied to the full dataset grayscale frequency histogram to generate a binary volume. The binarized volume was then used to calculate the LT of the segmented class (the gastrovascular system). In the dataset #1 (manubrium) two sub-datasets, the first featuring the scapulae and the second featuring the eight oral arms, were separated and analyzed independently. In the dataset #2 (the single oral arm) the three-winged portions were manually separated under the scapulae and then independently analyzed.

The binary volumes of the different systems were filtered using an isotropic Gaussian filter with a structuring element of five vx and followed by a segmentation preserving the original dataset connectivity to generate a smoother structure. This filtering is necessary to reduce the noise of the data and suppress the generation of many spurious branches in the skeletonization process. The 3D-isotropic-Gaussian filter fully preserves the mediality of the original dataset, while suppressing the smallest branches generated by the structure roughness during the thinning process. After the skeletonization, two consecutive four vx pruning cycles, to delete the remaining short ending branches, were applied. This combination of filtering and pruning operations provided the smallest number of spurious end branches (e.g., the ones radiating from the long oral arm structures, generated by changes in shape and not by the presence of canals), while preserving the short branches at the oral apertures. In order to provide a thickness-labeled skeleton, each vx in the resulting skeletons was labeled using the respective LT value. The analysis of the single oral arm was carried a step further, to also investigate the number of openings: the number of endpoints (ideally equivalent to the number of oral apertures) was then calculated, separately for each wing of one oral arm.

## Supporting information

**S1 Fig.** *Cotylorhiza tuberculata*, **interradial section of a specimen of about 15 cm in diameter.** Visible the Subgenital sinus (Sgs), two of the subgenital ostia (Asterisks), the stomach (St) with part of the gonads. Arrow indicates the area containing the complex of central branchings emerging from the 2-4-8 central canal system (schematized in S2 Fig.), including the future brood-carrying filaments and some smaller club-shaped digitations.
(TIF)

**S2 Fig.** *Cotylorhiza tuberculata,* **Schematic drawing of the central canal system at the base of the arm disc, immediately below the subgenital sinus.**
(TIF)

**S3 Fig.** *Cotylorhiza tuberculata,* **living specimen after about 1h after the injection (Methylene blue).** Exumbrellar view. Evident the stain present into the stomach, the per- and interradial canals (white and black arrows, respectively) and into the "pseudo ring canal complex" (See text). The web of adradial canals is now faded (red arrow). The outermost, distal part of the anastomoses was practically unstained, apart from a small area adjacent to the perradial canal on the left.
(IFF)

**S4 Fig.** *Cotylorhiza tuberculata,* **stain injection (Methylene blue).** Two living specimens, one at the moment of the injection into the stomach, the second after the injection.
(TIF)

**S1 Video. Micro CT-scan 3D rendering.** 360˚ animation of the manubrium endocast of the specimen of 15 cm in diameter.
(AVI)

**S2 Video. Micro CT-scan 3D rendering.** 360° animation of the endocast as above with the highlight (yellow) of the oral arm selected for the quantitative measurements, and the superimposed (red) segmented gastrovascular system.
(AVI)

**S3 Video. Micro CT-scan 3D rendering.** 360° animation of the endocast of the specimen of 35 cm in diameter.
(AVI)

**S4 Video. Toluidine blue stain diffusion into the gastrovascular system.** Specimen of 14 cm in diameter. Evident the coloration of the external portion of the oral arm canals and the branches of the dichotomized distal portions. About 10 min after injection.
(MP4)

**S5 Video.** *Cotylorhiza tuberculata.* Adult specimen of about 30 cm in diameter, swimming in the Gulf of Trieste, clip taken on August 25, 2022. The solidity of the central portion of the umbrella is evident. The central dome is immobile, the subsequent crown (about 1/3–2/3 of the diameter) flexes slightly during pulsation. The only truly flexible portion is the distal part.
(MP4)

## Author contributions

**Conceptualization:** Gregorio Motta, Marco Voltolini, Lucia Mancini, Diego Dreossi, Massimo Avian.

**Data curation:** Marco Voltolini.

**Formal analysis:** Gregorio Motta, Marco Voltolini.

**Funding acquisition:** Antonio Terlizzi, Massimo Avian.

**Investigation:** Gregorio Motta, Marco Voltolini, Lucia Mancini, Diego Dreossi, Valentina Tirelli, Lorenzo Peter Castelletto, Manja Rogelja, Massimo Avian.

**Methodology:** Gregorio Motta, Marco Voltolini, Lucia Mancini, Diego Dreossi, Francesco Brun.

**Project administration:** Antonio Terlizzi, Massimo Avian.

**Supervision:** Antonio Terlizzi, Massimo Avian.

**Validation:** Massimo Avian.

**Visualization:** Gregorio Motta, Marco Voltolini, Massimo Avian.

**Writing – original draft:** Gregorio Motta, Marco Voltolini, Diego Dreossi, Massimo Avian.

**Writing – review & editing:** Gregorio Motta, Marco Voltolini, Lucia Mancini, Diego Dreossi, Francesco Brun, Valentina Tirelli, Manja Rogelja, Antonio Terlizzi, Massimo Avian.

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
