## [Decision Letter · Decision Letter 0]

19 Jun 2025

Dear Dr. Motta,

Thank you for submitting your manuscript to PLOS ONE. After careful consideration, we feel that it has merit but does not fully meet PLOS ONE’s publication criteria as it currently stands. Therefore, we invite you to submit a revised version of the manuscript that addresses the points raised during the review process.

We look forward to receiving your revised manuscript.

Kind regards,

Phuping Sucharitakul

Academic Editor

PLOS ONE

3. We note that Figure 12 in your submission contain [map/satellite] images which may be copyrighted. All PLOS content is published under the Creative Commons Attribution License (CC BY 4.0), which means that the manuscript, images, and Supporting Information files will be freely available online, and any third party is permitted to access, download, copy, distribute, and use these materials in any way, even commercially, with proper attribution. For these reasons, we cannot publish previously copyrighted maps or satellite images created using proprietary data, such as Google software (Google Maps, Street View, and Earth). For more information, see our copyright guidelines: http://journals.plos.org/plosone/s/licenses-and-copyright.

1. You may seek permission from the original copyright holder of Figure(s) [#] to publish the content specifically under the CC BY 4.0 license.   

Additional Editor Comments (if provided):

Reviewers' comments:

Reviewer's Responses to Questions

**Comments to the Author**

1. Is the manuscript technically sound, and do the data support the conclusions?

Reviewer #1: Yes

2. Has the statistical analysis been performed appropriately and rigorously?

Reviewer #1: N/A

3. Have the authors made all data underlying the findings in their manuscript fully available?

Reviewer #1: Yes

4. Is the manuscript presented in an intelligible fashion and written in standard English?

Reviewer #1: Yes

Reviewer #1: First, I’d like to congratulate the authors on the work. This is a very interesting study which can certainly be useful for enriching future taxonomic descriptions in Cnidaria with more anatomical details as well as revise old descriptions, and contribute to studies on physiology and evolution in Animalia.

The manuscript is generally well written, all the goals were achieved, and figures and supplementary material are informative and helpful.

Overall, I recommend the publication of this study. However, some corrections and additional information must be added. I’m sending the PDF with my suggestions attached.

Best wishes,

Hellen

**Do you want your identity to be public for this peer review?** For information about this choice, including consent withdrawal, please see our Privacy Policy

Reviewer #1: **Yes: ** Hellen Ceriello

---

## [Author Response · Author response to Decision Letter 1]

4 Jul 2025

Editor Comments to the Author.

Dear editor, we provide below our answers to the journal requirements.

R1: Dear editor, we checked all the style requirements, and we formatted the text as required.

R2: Dear editor, no permit to access sampling sites was required because the areas are part of the public maritime domain. We added a sentence at lines 547-548.

3. We note that Figure 12 in your submission contain [map/satellite] images which may be copyrighted. All PLOS content is published under the Creative Commons Attribution License (CC BY 4.0), which means that the manuscript, images, and Supporting Information files will be freely available online, and any third party is permitted to access, download, copy, distribute, and use these materials in any way, even commercially, with proper attribution. For these reasons, we cannot publish previously copyrighted maps or satellite images created using proprietary data, such as Google software (Google Maps, Street View, and Earth). For more information, see our copyright guidelines: http://journals.plos.org/plosone/s/licenses-and-copyright.

R3: Dear Editor, for Figure 12, the map source is Natural Earth (Free vector and raster map data @naturalearthdata.com), which material are in the public domain, no permission to use is needed. We also uploaded the additional “other” proof file that better explains their use declaration.

R4: Dear editor, we checked the bibliography, and it is complete and correct. We did not add or remove any citation.

Review Comments to the Author.

Reviewer #1: First, I’d like to congratulate the authors on the work. This is a very interesting study which can certainly be useful for enriching future taxonomic descriptions in Cnidaria with more anatomical details as well as revise old descriptions and contribute to studies on physiology and evolution in Animalia. The manuscript is generally well written, all the goals were achieved, and figures and supplementary material are informative and helpful. Overall, I recommend the publication of this study. However, some corrections and additional information must be added. I’m sending the PDF with my suggestions attached.

Best wishes,

Hellen

R: Dear Hellen, thanks for your appreciation, comments, and suggestions, which have helped us improve the quality of our work. A detailed reply to your specific comments is provided below.

Comment 1: Keywords should be different from the ones present in the title. Suggestion: Cnidaria, CT scan, morphology

R1: Dear reviewer, thanks for the comment. We agree that keywords should not match the title. We decided to keep those you suggested plus “jellyfish physiology; digestive system evolution” which are not mentioned in the title.

Comment 2: Line 58. Maybe replace with "bypass" or "overcome"?

R2: Thanks for the suggestion, we changed the term to overcome at line 58.

Comment 3: Line 65. Is there an extra space between these 2 words?

R3: We corrected and deleted the extra space at line 65.

Comment 4: Line 71. widely?

R4: Thanks for the suggestion, we changed to widely at line 71.

Comment 5: Line 74. Authority and year of publication should be mentioned here and deleted from Methodology section.

R5: We agree with the reviewer. We added the informationat lines 75-76.

Comment 6: Line 77. in*

R6: Thanks for noticing the error, we corrected at line 78.

Comment 7: Line 81. Authority and year of publication

R7: We agree with the reviewer. We added the informationat line 82.

Comment 8: Line 109. Indicated by the red arrows in Fig 2B?

Comment 9: Line 119. The figure caption informs "oral arm canal".

Comment 10: Line 109. Are those the "scar" mentioned in the figure caption? It doesn't sound clear to me.

Comment 11-12: Line 116. The caption is clear, but it contrasts with the text itself. It's confusing. Please, make the text (Lines 108-114) clearer to match the caption.

R8-12: Dear reviewer, thank you for your comments. We have revised the text to improve overall clarity. In particular, we added a bracket at line 110 to better define the fact that the black arrow of figure 2 indicates the Y-shaped relief structure and, accordingly, we changed the term “scar” in the figure legend (line 121) so that it now matches with the text. Plus, we added reference to the red arrows at lines 112-113 (that indicates the oral arm canals) and a white arrow with a reference in the text (line 111 and 122) to highlight the bifurcations.

We hope these changes made the text easier to read and improved the correspondence between the text and the figure legend.

Comment 13: Line 160-161. That's tricky because the counting may vary depending on the observer.

R13:

Dear reviewer, we know that counting canals may be affected by the observers’ subjectivity. For this reason, we considered counting only the openings on the margin of the stomach the most objective strategy. Attempting to count potential dichotomies (especially those that begin close to the margin) would introduce greater variability and interpretative ambiguity. Therefore, our approach should reduce observer bias and ensure consistency in the analysis.

Comment 14: Line 176. Delete

R14: We decided to keep the reference to figure 2B since it indicates the canals (red arrows) and we removed Figure 4B at line 179.

Comment 15: Line 180. For better visualization could you color both arrows white?

Comment 16: Line 181. I don't see why this is relevant. Maybe delete B.

R15-16: Dear reviewer, we agree with the suggestion and we deleted figure 4B. We also changed the arrow color in figure 4A.

Comment 17: Line 190. approximately?

Comment 18: Line 190. Please, write in full.

R17-18: We added the information (1 up to 2 cm) at line 193.

Comment 19: Line 190. Delete?

R19: Deleted the reference to figure 4A and 4B (now it is just Figure 4) at line 193.

Comment 20: Line 198. C as in B as in A... it's confusing. Please, be clearer.

R20: We agree with the reviewer about the complexity of the original caption. We have revised the whole Fig 6 caption, and we hope that clarity has been significantly improved.

Comment 21: Line 211. Please, color the "A" in the figure white

R21: Changed the color.

Comment 22: Line 212. Suggestion: 1st, 2nd. 3rd

R22: Changed as suggested at line 222.

Comment 23: Line 226. approximately?

R23: We added the info (up to 5 cm in larger specimens) at line 236.

Comment 24: Line 228. Delete

R24: Dear reviewer, we have decided to keep both Figure 5A and 5B, as they respectively show the resin cast and the micro-CT scan. This allows for a direct visual comparison, highlighting the high fidelity of the scan in reproducing even the smallest features of the original sample.

Comment 25: Line 245. This is not clear

R25: Thank you for the comment. We changed the caption of figure 8 at line 255, we hope that now it is clearer.

Comment 26: Line 263-264. Suggestion: [...] analysis where all canal terminations were considered endpoints.

R26: We agree with the comment, we changed the text as suggested at line 274-276.

Comment 27: Line 279. Total number of branches

Comment 28: Line 279. Total number of junctions

Comment 29: Line 279. Total number of endpoints

R27-29: We agree with the comments, and we changed the text as suggested in Table 1.

Comment 30: Line 301. Delete?

R30: Dear Reviewer, thank you for your suggestion. We prefer to keep Figure 10B despite its similarity to Figure 10A. The reason is that Figure 10A shows both canals in which the dye is present due to internal circulation and an example of an injection artifact (indicated by the asterisk). In contrast, Figure 10B illustrates canals where the dye has passively diffused, and it also shows the stomach's appearance after the injection.

Comment 31: Line 340. that we*

R31: Changed as suggested at line 352.

Comment 32: Line 342. Delete

R32: Changed as suggested at line 354.

Comment 33: Line 343. spreaded*

R33: Changed to past form as suggested at line 355.

Comment 34: Line 345. Delete

R34: Changed as suggested at line 357.

Comment 35: Line 346. began*

R35: Changed as suggested at line 358.

Comment 36: Line 346. became*

R36: Changed as suggested at line 358.

Comment 37: Line 347. was

R37: Changed as suggested at line 359.

Comment 38: Line 348. required

R38: Changed as suggested at line 360.

Comment 39: Line 348. longer*

R39: Changed as suggested at line 360.

Comment 40: Line 353. had

R40: Changed as suggested at line 365.

Comment 41: Line 362-363. Please, indicate it in the image. There is no asterisk in C.

R41: Dear reviewer, we noticed that we erroneously labeled figure 11C (now 11D) and 11D (now 11C). Now the caption is correct.

Comment 42: Line 433-434. Authority and year of publication

Comment 43: Line 434. Authority and year of publication

R42-43: Thanks for the suggestions, we added both information at lines 446-447.

Comment 44: Line 487. oral*

R44: Corrected at lines 499-500.

Comment 45: Line 520. Suggestion: Lobonemoides gracilis Light, 1914 (previously known as L. robustus).

R45: Changed as suggested at lines 532-533.

Comment 46: Line 536. Suggestion: along with the one by Avian and colleagues

R46: Changed as suggested at line 549.

Comment 47: Line 537. Delete

R47: Changed as suggested at line 550.

Comment 48: Line 542. Delete

R48: Changed as suggested at line 555.

Comment 49: Line 542. Please, include the total number of specimens.

R49: Thanks for the comment, we added the number of specimens at line 555.

Comment 50: Line 543. Fig 1 does not show location info. Instead, it shows the jellyfish. I suggest removing it from here and place in line 542 after "individuals".

Comment 51: Line 544. Fig 12 should be mentioned here.

R50-51: Dear reviewer, thanks for the comment. We wrote Figure 1 instead of Figure 12. We corrected the text at line 557-558.

Comment 52: Line 546. I found the map and caption confusing. One image is overtopping the other and there are too many shapes (circle, triangles, square) and colors (green, blue, yellow, red) that are not well explained. Please, simplify it.

R52: Dear reviewer, thank you for your comment. We provided a new Figure 12 caption that now explains every detail and we hope it is now less confusing.

Comment 53: Line 549. Length or diameter? Please, specify it in the caption.

R53: We added the information in the table header.

Comment 54: Line 549. Please, standardize the use of – (en dash) or - (dash).

R54: We changed the text to classic dash – in table 2.

Comment 55: Line 553. different sizes

R55: Corrected as suggested at line 575.

Comment 56: Line 555. Table 2 informs up to 20 cm.

R56: Thank you for the comment. We have now changed the size at line 577.

Comment 57: Line 556. (Table 2)

R57: Added the reference at line 578.

Comment 58: Line 557. What about the 24 juvelines from 2023?

R58: Dear reviewer, we have now added the info about 2023 individuals at lines 580-581.

Comment 59: Line 557. How many specimens per tank?

R59: Info added at line 579.

Comment 60: Line 563. How many mL each?

R60: Dear Reviewer, the injection volume depended on the size of the individual: larger animals required more dye. However, care was taken to avoid over-injection, as the goal was to fill the stomach adequately without exceeding its capacity. Overfilling could cause the dye to be pushed into underlying canals due to injection pressure rather than by the jellyfish's internal currents. We added the mL info at lines 587-589.

Comment 61: Line 565. How is that possible if first sampling occurred in 20 July 2022 (Table 2)?

R61: Thanks for the comment, we reported the wrong date. We corrected the sampling days in Table 2.

Comment 62: Line 583. Out of curiosity, how much does this approach cost (contrast injections + resin casts + X-ray)?

Comment 63: Discussing the cost-benefit of this approach seems rather relevant.

R62-63: Dear Reviewer, thank you for your comment. We already have reported the brand of the resin we used so it can be found on the market (it is not expensive, less than 50 Euro/L). Similarly, the stains we used are common in laboratories and do not represent a significant expense.

We did not include the cost of the scanning because there is no standardized or universally applicable pricing currently available on the market. The final cost strongly depends on several variables, including the size of the sample and the resolution required, both of which significantly influence the scanning time and, consequently, the price. In our case, we conducted the scans within the framework of a scientific collaboration, which granted us a significant discount compared to commercial/private rates. As such, the cost we incurred would not be representative or informative for other institutions, especially considering the potential variability across countries and service providers. For these reasons, we chose not to report a specific price and to discuss the costs-benefits since it would be potentially misleading and not valid worldwide.

Comment 64: Line 592. mm

R64: Corrected at line 617.

Comment 65: Line 593-594. Suggestion: an increase of approximately 60% in the lateral...

R65: Dear reviewer, this technical section was written by the physicians who performed the scan, so we prefer to retain their terminology and syntax.

Comment 66: Line 603. N

R66: Corrected at line 628.

Comment 67: Line 612. N

R67: Corrected at line 637.

Comment 68: Line 657. Delete

R68: Corrected at line 682.

Comment 69: Line 659. Then independently?

R69: Corrected at line 684.

Comment 70: Line 662. Delete

R70: Corrected at line 687.

Comment 71: Line 667. Delete

R71: Corrected at line 692.

---

## [Decision Letter · Decision Letter 1]

5 Aug 2025

Dear Dr. Motta,

Thank you for submitting your manuscript to PLOS ONE. After careful consideration, I am pleased to inform you that the manuscript looks satisfactory and will be accepted pending minor revisions.

We look forward to receiving your revised manuscript.

Kind regards,

Phuping Sucharitakul

Academic Editor

PLOS ONE

Journal Requirements:

Additional Editor Comments (if provided):

I am pleased to inform you that the manuscript looks satisfactory and will be accepted pending minor revisions.

Reviewers' comments:

Reviewer's Responses to Questions

**Comments to the Author**

Reviewer #1: All comments have been addressed

Reviewer #2: (No Response)

Reviewer #3: All comments have been addressed

2. Is the manuscript technically sound, and do the data support the conclusions?

Reviewer #1: Yes

Reviewer #2: Partly

Reviewer #3: Yes

3. Has the statistical analysis been performed appropriately and rigorously?

Reviewer #1: N/A

Reviewer #2: N/A

Reviewer #3: N/A

4. Have the authors made all data underlying the findings in their manuscript fully available?

Reviewer #1: Yes

Reviewer #2: Yes

Reviewer #3: Yes

5. Is the manuscript presented in an intelligible fashion and written in standard English?

Reviewer #1: Yes

Reviewer #2: Yes

Reviewer #3: Yes

Reviewer #1: (No Response)

Reviewer #2: The manuscript addresses interesting questions and shows how they can be answered by new modern methods and approaches. Obtained results can be of interest not only for the specialists working with scyphozoans. Most of the illustrations are of high quality and informative.

Unfortunately, the manuscript is not without certain drawback. Firstly, it is too diffuse (verbose). Secondly, some topic in discussion are not directly connected with the topic of the work and sounds fondly. Thirdly, some illustrations dublicate one another, or redundant (give no additional information).

I recommend minor revision.

Following are the certain remarks for the text:

Line 22 - …”jellyfish”... – jellyfish anatomy.

Line 28 - …“adding numerous details”… - you mention only one detail…

Lines 32-33 – “These findings challenge the theory of a simple digestive system in cnidarians” - What are the doubts? That the system is “simple”? The question is in the definition of “simple system”…

Line 33 – “Given the genetic distance between Cotylorhiza tuberculata and Rhizostoma pulmo,”… - With what to compare this distance? And how it affect the organisation of the digestive system?

Line 35 – “…suggesting that jellyfish are more complex organisms than previously thought.” - In what sense are they more complex? The plan of Cnidaria organization remains unchanged.

Line 56 – “…proper “internal” description of anatomy…” – proper description of “internal” anatomy.

Line 63 – “ and the classic bibliography…” - What does the bibliography have to do with it?

Lines 68-69 – “…Cnidaria and Ctenophora have simple, single-cavity digestive systems [23], while the first through-gut appeared in Bilateria…” – it is a comparison of ‘square and round’.

Lines 69-71 – the last sentence of the paragraph - A fully far-fetched topic. As many cnidarians have two-directional flows in their digestive system.

Lines 79-80 – “…as it is widely distributed for both its abundance and distribution in the Mediterranean Sea…: - How is that?

Line 80 – see comments to line 33.

Lines 85-86 – “Our aim was to… (2) describe the anatomy and functional anatomy of this species in three dimensions and great detail…” – there are a lot more details visible in illustrations compared to the discussed in the text.

Lines 131-137 – May be it is better to move this paragraph to the material and Methods.

Lines 144-146 – the last sentence “This suggests that the subgenital…” it is discussion…

Line 164 – “flattened at the ex-subumbrellar plane” – along oral-aboral axis.

Lines 166-167 – “which have no connections” – no connection to what?

Lines 168-169 – “…which then decreases in the last distal third…” - one canal decreases, the other one – increases after decreasing…

Lines 169-171 – “According to the current state of the art, the umbrellar canals pattern of C. tuberculata is described as having no annular (or ring) canals.” – It is discussion + it looks like as wide canal with points of connection between floor and roof walls.

Line 173 – “…larger, perpendicular canal (connected to the anastomosis network)…” – as a part of anastomosis network.

Lines 174-175 – “Some zooxanthellae clusters … of the canals” – please, indicate at the Figure.

Line 179 – “…slight terminal dilatations…” which dilations – it is not clear from the figure.

Lines 208-212, 224, 236 – Can it be an artefact? Adhesion due to fixation?

Lines 256-261 – move to Material and Metods.

Lines 300-301 – “…affects only the most lateral, slightly dilated part of the oral arm canals (Fig 10A-10C; S4 and S5 Video).” – is not convincing arguments – Fig. 10A – it is Ok, but on D-C and in Videos – not clear. The canal borders are not clearly distinguishable, and in Video the stain moves first in the center of the canal. Can it depend on the amount of fluid (food)?

Lines – 318- 324 – move to the Material and methods.

Lines 327-331 – it is discussion.

Lines 338-339 – “…just before the bending point of the dye never…” – some words were lost.

Line 346 – “…and spread towards the inter- and perradial canals.” – it should be shown.

Lines 372-372. The first sentence sounds contradictory in relation to the introduction, which said that there were few works only.

Lines 410-412 – “It is logical to hypothesize … the extension of four other invaginations of the mesoglea into the basal pillars ….resulting in the dichotomies present in the scars ” – what is the mechanism of mesoglea extension? The result is clear, but the mechanism is farfetched.

Lines 428-429 – reference to the Fig 11D, 11E; S3 Fig is improbable.

Lines 430-435 - childish prattle… The authors wrote themselves, that it would be counted according to the number of openings…

Lines 436-438 – “…as the differences within the genus Cotylorhiza are really minimal. …” - the difference is sufficient, especially the scatters of the values do not overlap.

Lines 440- 442 – “These strips look more wrinkled … and outer parts of the canals (Fig…” - The assumption is possible, but it sounds strained, requires proofing.

Lines 442-443 – “which is also evident from the flow simulation (Fig 7C).” - The model cannot serve as proof, as it works according to the laid program. Can help in forecasting.

Lines 461-462 – “This adaptation could also justify the reduced number of openings present on the oral arms.” - the correlation is not evident…

Lines 462-467 – The last sentences of the paragraph have no connection with the work.

Lines 473-474 – “as commonly observed in other jellyfish” – it is not common for all jellyfishes - completely different situation in Aurelia.

Lines 476-477 – “…was never observed (in a few cases, only some octants per jellyfish were totally filled)…” – finally: was never observed or was observed in few cases?

Line 494 – reference to figure 11B - is poorly visible at the figure.

Lines 502-507 – the whole paragraph is doubling of the previous discussions… The reasoning is very verbose, with repetitions.

Lines 513-515 – the same…

Lines 524-526 - What does it have to do with the topic of the work? Far-fetched ideas... Have you ever looked at actinia or corals, how they ventilate their gastral cavity?

Lines 531-538 – the whole paragraph is absolutely nothing. Primitive idea. No considerations about physiology. As a hydra at school learned, so everything measured from this point of view.

Lines 858 – “…some minutes from the injection…” – after the injection.

Line 867 – “Evident the coloration of the external portion of the oral arm canals…” – not clear, especially in the upper part.

Fig. 1 – it worth replacing this beautiful collage with the jellyfish photo and scheme of organization.

Fig. 2. “(A) Dissection” - transverse dissection.

Line 121 – “…white arrow indicates…” – the white arrow is visible well only after extension of the image… Maybe it is better to change its color.

Fig 3. Line 154 – “canals” – canal.

(B) “Detail of (A), showing a pair of adradial canals (arrows) with a single origin (asterisk)” – inconclusive (unconvincing) example, it is better visible on left side of A.

Line 157 – “The square brackets show the widening of some canals…” – widening and fusion…

Fig. 5. Line 187 – “the” – The…

Fig. 6. Lines199-200 – “Section of another oral arm from the specimen shown in (B), with the lateral side on the left. …” – there is no area of adhesion...

Fig. 10. Line 310-311 – “The arrows indicate the staining of one of the inner part of an oral arm canal (artifact) …” – why it is an artifact?

Line 313 – “the” – The…

Fig. 11. There is reference in the txt only to Fig. 11B, no more. Moreover, doubling of fig. 3 and 10…

Fig. 12. – large map, but what new information does it contain? Can be replaced by an inset somewhere.

S1 Fig – lines 847-848 – “Arrow indicates the complex of central branchings emerging from the 2-4-8 central canal system, …”- not clear. It worth replacing with a scheme of organization.

S5 Video – beautiful, but gives no additional information except canal dichotomy.

S6 Video - the same situation – what did authors want to show?

Reviewer #3: The manuscript has already been evaluated. The authors provided adequate responses to all comments and suggestions, following most of them and presenting reasons when do not agree completely with the reviewer.

This manuscript advances the knowledge of jellyfish anatomy using modern techniques and provide substantial data and information to help in the systematics of the group and physiology of the digestive system of rhizostome jellyfishes.

In my opinion it is fully accepted for publication.

**Do you want your identity to be public for this peer review?** For information about this choice, including consent withdrawal, please see our Privacy Policy

Reviewer #1: **Yes: ** Hellen Ceriello

Reviewer #2: **Yes: ** Kosevich Igor A.

Reviewer #3: **Yes: ** Andre C. Morandini

---

## [Author Response · Author response to Decision Letter 2]

6 Oct 2025

Dear Editor,

We greatly acknowledge the efforts of the reviewers and the editor. All comments were carefully considered, and the manuscript was revised accordingly. As requested, we uploaded both the track-change manuscript and unmarked version. Here enclosed please find our reply to the comments from the reviewers in point-by-point fashion.

Kind regards,

Gregorio Motta

Response to comments of Reviewer #2

General comment:

Reviewer #2: The manuscript addresses interesting questions and shows how they can be answered by new modern methods and approaches. Obtained results can be of interest not only for the specialists working with scyphozoans. Most of the illustrations are of high quality and informative.

Unfortunately, the manuscript is not without certain drawback. Firstly, it is too diffuse (verbose). Secondly, some topic in discussion are not directly connected with the topic of the work and sounds fondly. Thirdly, some illustrations duplicate one another, or redundant (give no additional information).

I recommend minor revision.

Response: Dear reviewer, thank you very much for your appreciation and valuable suggestions. We have removed the redundant parts, and we hope the manuscript is now more concise. A detailed point-by-point reply to your specific comments is provided here below.

Following are the certain remarks for the text:

Comment #1: Line 22 - ..."jellyfish"... – jellyfish anatomy.

Response: We agree with the suggestion and we changed the text accordingly at line 22.

Comment #2: Line 28 - ..."adding numerous details"... - you mention only one detail...

Response: Dear reviewer, thanks for the comment. We decided to change the text at lines 27-30 to better define the fact that after the comma we put the focus on one of the various details.

Comment #3: Lines 32-33 – "These findings challenge the theory of a simple digestive system in cnidarians" - What are the doubts? That the system is "simple"? The question is in the definition of "simple system"...

Response: We agree with the comment. We better explained that we consider simple the traditional scheme of scyphozoan oral openings with both intake-outtake functions compared to the specialized openings we found. We added a sentence at lines 34-35 and we hope this helps to better clarify.

Comment #4: Line 33 – "Given the genetic distance between Cotylorhiza tuberculata and Rhizostoma pulmo,"... - With what to compare this distance? And how it affect the organization of the digestive system?

Line 35 – "...suggesting that jellyfish are more complex organisms than previously thought." - In what sense are they more complex? The plan of Cnidaria organization remains unchanged.

Response: Thanks for the useful comment. We understand that we have not been sufficiently clear. Here we want to stress the fact that C. tuberculata and R. pulmo belong to different suborders (thus not so near genetically speaking inside scyphozoans) but they share this similar feature, the specialized openings, which have never been described before in other jellyfish. Therefore, we suggest that maybe this anatomical characteristic may be more diffused inside the order Rhizostomeae than generally thought. We added a clarifying sentence at lines 36-39 and removed the last sentence to tone it down.

Comment #5: Line 56 – "...proper "internal" description of anatomy..." – proper description of "internal" anatomy.

Response: Corrected at line 60 as suggested.

Comment #6: Line 63 – " and the classic bibliography..." - What does the bibliography have to do with it?

Response: We agree with your comment. We changed the term bibliography, which was not the correct word, with descriptions at line 67.

Comment #7: Lines 68-69 – "...Cnidaria and Ctenophora have simple, single-cavity digestive systems [23], while the first through-gut appeared in Bilateria..." – it is a comparison of 'square and round'.

Response: Thanks for the comment. We have modified the text from line 71 to 77. We hope it is now clearer that we considered cnidarians and ctenophores together (although we are aware that they represent rather different groups), but in this context they share the feature of being positioned before bilaterians in the evolutionary tree. Importantly, in both groups, some species were demonstrated to possess more complex digestive systems (with distinct openings dedicated to ingestion or egestion) than the common single cavity.

Comment #8: Lines 69-71 – the last sentence of the paragraph - A fully far-fetched topic. As many cnidarians have two-directional flows in their digestive system.

Response: We agree with the reviewer, we removed lines 75-77.

Comment #9: Lines 79-80 – "...as it is widely distributed for both its abundance and distribution in the Mediterranean Sea...: - How is that?

Response: We noticed the error in the text and we changed it at line 85.

Comment #10: Line 80 – see comments to line 33.

Response: Dear reviewer, as for the comment at line 33 we added the reference at line 86-87 of their genetic distance to better clarify that we choose C. tuberculata since it is an important species in the Mediterranean Sea (and this also helps with finding enough specimen to analyze) and because this species and R. pulmo (which we already analyzed with cast and microCT) are not so closely related between Rhizostomeae, thus giving the opportunity to assess the potential differences and similarities inside the order Rhizostomeae.

Comment #11: Lines 85-86 – "Our aim was to... (2) describe the anatomy and functional anatomy of this species in three dimensions and great detail..." – there are a lot more details visible in illustrations compared to the discussed in the text.

Response: We understand your point. We rephrased the objective sentence for better clarity at lines 91-93 and we removed lines 93-97.

Comment #12: Lines 131-137 – May be it is better to move this paragraph to the material and Methods.

Comment #13: Lines 144-146 – the last sentence "This suggests that the subgenital..." it is discussion...

Response: We agree with the reviewer for both suggestions. We moved and rearranged the first one in the materials and methods at lines 688-694 and the second in the discussions at lines 452-454.

Comment #14: Line 164 – "flattened at the ex-subumbrellar plane" – along oral-aboral axis.

Response: We thank the reviewer for this correction, we changed the sentence to oral-aboral axis at line 173.

Comment #15: Lines 166-167 – "which have no connections" – no connection to what?

Response: We changed the sentence at line 176. We hope that now it is clearer that adradial and perradial canals have a lot of anastomoses along their entire length, while interradial near their origin have no anastomoses-connections with other canals.

Comment #16: Lines 168-169 – "...which then decreases in the last distal third..." - one canal decreases, the other one – increases after decreasing...

Response: Dear Reviewer, we understand that the terminal part of the canal may appear to widen; however, this portion actually corresponds to the region of the rhopalium. For this reason, we consider it more accurate to state that the canal itself ends at the narrowest point and does not expand further.

Comment #17: Lines 169-171 – "According to the current state of the art, the umbrellar canals pattern of C. tuberculata is described as having no annular (or ring) canals." – It is discussion + it looks like as wide canal with points of connection between floor and roof walls.

Response: We agree with the reviewer, we removed the sentence and we rephrased the paragraph in the discussions at lines 490-494.

Comment #18: Line 173 – "...larger, perpendicular canal (connected to the anastomosis network)..." – as a part of anastomosis network.

Response: Changed as suggested at line 182.

Comment #19: Lines 174-175 – "Some zooxanthellae clusters ... of the canals" – please, indicate at the Figure.

Response: We added a reference to figure 3A-B and a sentence about zooxanthellae at line 167 in figure 3 caption for better clarity.

Comment #20: Line 179 – "...slight terminal dilatations..." which dilations – it is not clear from the figure.

Response: We agree with the comment. We added the reference to the white arrows at line 189.

Comment #21: Lines 208-212, 224, 236 – Can it be an artefact? Adhesion due to fixation?

Response: Dear reviewer, we observed this adhesion (now changed to apposition since it is a better anatomical terminology for two tissues which are in close contact, often touching, but not fused or stuck together) in all our experiments both on fresh tissues of jellyfish and fixed samples, so we can reject the hypothesis of the artifact.

Comment #22: Lines 256-261 – move to Material and Methods.

Response: We agree and we thank the reviewer for the suggestion, we decided to remove the paragraph since it would be redundant in the materials and methods.

Comment #23: Lines 300-301 – "...affects only the most lateral, slightly dilated part of the oral arm canals (Fig 10A-10C; S4 and S5 Video)." – is not convincing arguments – Fig. 10A – it is Ok, but on D-C and in Videos – not clear. The canal borders are not clearly distinguishable, and in Video the stain moves first in the center of the canal. Can it depend on the amount of fluid (food)?

Response: Dear Reviewer, thank you for your valuable comment. As also mentioned in response to other remarks, we have revised the manuscript by adding Figure 11 illustrating the anatomical scheme of the canals in the oral arms, in order to facilitate the interpretation of the images of the internal circulation. The canal emphasized in these images corresponds to the upper/external canal, which is responsible for egestion, whereas in the lower/internal part of the oral arm—here completely transparent—the other canal is located, which is responsible for ingestion.

Comment #24: Lines – 318- 324 – move to the Material and methods.

Response: We agree with the reviewer, we moved the paragraph to lines 663-668 in the materials and methods.

Comment #25: Lines 327-331 – it is discussion.

Response: Dear reviewer, we prefer to leave this paragraph here as it provides a descriptive account of the diffusion of the stain without entering into interpretation or discussion. We believe it contributes useful context for the reader.

Comment #26: Lines 338-339 – "...just before the bending point of the dye never..." – some words were lost.

Response: Thanks for the comment, we corrected the sentence at line 388-392.

Comment #27: Line 346 – "...and spread towards the inter- and perradial canals." – it should be shown.

Response: We agree with the comment; we put better references for Figures at line 399. We also added arrows in Figure S3 and modified the figure S3 caption to better explain the circulation.

Comment #28: The first sentence sounds contradictory in relation to the introduction, which said that there were few works only.

Response: We understand the point made by the reviewer, the terms historically and traditional could misleadingly suggest that this research field was more extensively investigated and has a larger scientific community than it actually was. We changed the first sentence of the discussion at line 426, and we hope that know it is more clear.

Comment #29: Lines 410-412 – "It is logical to hypothesize ... the extension of four other invaginations of the mesoglea into the basal pillars ....resulting in the dichotomies present in the scars " – what is the mechanism of mesoglea extension? The result is clear, but the mechanism is farfetched.

Response: Dear reviewer, this paragraph refers to the work of Avian, 1986. In that case, it was possible to follow the development up to the 1-cm diameter meta-ephyra stage. One of us (Avian M.) rewrote this section to clarify the hypothetical subsequent, unfortunately undocumented development.

We changed the whole paragraph from line 462 to 469.

Comment #30: Lines 428-429 – reference to the Fig 11D, 11E; S3 Fig is improbable.

Response: We noticed the error in the reference to figures and we changed it at line 494.

Comment #31: Lines 430-435 - childish prattle... The authors wrote themselves, that it would be counted according to the number of openings...

Comment #32: Lines 436-438 – "...as the differences within the genus Cotylorhiza are really minimal. ..." - the difference is sufficient, especially the scatters of the values do not overlap.

Response: Dear reviewer, although the question may seem obvious, it really has some importance, given that the number and shape of the umbrellar canals are considered diagnostic characters in virtually all jellyfish identification guides and related dichotomic keys. Indeed, it is common practice to detect this characteristic simply by observing and counting canals by transparency in the mid-distal part of the umbrella (without properly checking the origins of such canals), with the risk of overestimation. We changed the paragraph from line 503 to 516 and we hope now the concept is clearer.

Regarding the potential overlap, we respectfully maintain our interpretation: we observed 7–11 canals per octant, which show partial correspondence with both C. erythraea (4–8 adradial canals per octant) and C. ambulacrata Haeckel, 1880 (11–13 adradial canals) features.

Comment #33: Lines 440- 442 – "These strips look more wrinkled ... and outer parts of the canals (Fig..." - The assumption is possible, but it sounds strained, requires proofing.

Response: Dear reviewer, we think that it can be an issue with the scarce quality of the upload before the proofs. In Figure 6 at high quality the two tissues in the apposition part are not as smooth as the same tissues when they are more distant from each other, and the wrinkled part is more evident.

Comment #34: Lines 442-443 – "which is also evident from the flow simulation (Fig 7C)." - The model cannot serve as proof, as it works according to the laid program. Can help in forecasting.

Response: We understand the comment made by the reviewer. We found the separation of flows is not completely sealed in the experiments with live specimens in which we observed that sometimes the stain diffused from one canal to the other (medial part in Fig 6B). Other specimens, used for casts, showed the same feature, highlighted by the circulation model. Thus, we know that the circulation model (made from the cast scan) alone would not serve as proof but, in our case, it confirms what was observed in dissected samples.

Comment #35: Lines 461-462 – "This adaptation could also justify the reduced number of openings present on the oral arms." - the correlation is not evident...

Response: Dear reviewer, we agree with your comment, and we rephrased the paragraph at line 536-540 to lower the tone. We do not want to provide an explanation, but it is just our assumption. Since the Rhizostoma specimens we analyzed were not fully mature but possessed a higher number of mouths than Cotylorhiza, which, having fewer, has a lower potential energetic intake yet is able to grow more rapidly, we hypothesized that a substantial proportion of its energetic needs derives from the zooxanthellae.

Comment #36: Lines 462-467 – The last sentences of the paragraph have no connection with the work.

Response: As suggested, we removed the whole paragraph.

Comment #37: Lines 473-474 – "as commonly observed in other jellyfish" – it is not common for all jellyfishes - completely different situation in Aurelia.

Response: Dear reviewer, as reported in Russell (1970, The Medusae of the British Isles volume II: pelagic scyphozoa, with a supplement to the first volume of Hydromedusae), who described the internal circulation in Aurelia aurita (scheme at page 159) which is still considered as valid nowadays, the centrifugal circulation runs along the adradial canals and the centripetal along the per and interradial canals. Here, in C. tuberculata, we described the same pattern. However, for better clarity and focus, we changed the sentence “as commonly observed in other jellyfish" to “as commonly observed in other scyphozoan jellyfish" and we corrected the sentence from line 553-555.

Comment #38: Lines 476-477 – "...was never observed (in a fe

---

## [Editor Report · Decision Letter 2]

30 Oct 2025

New advances in jellyfish anatomy: the benefits of endocasts and X-ray microtomography in the investigation of the gastrovascular system of Cotylorhiza tuberculata (Scyphozoa; Rhizostomeae; Cepheidae).

PONE-D-25-22492R2

Dear Dr. Motta,

We’re pleased to inform you that your manuscript has been judged scientifically suitable for publication and will be formally accepted for publication once it meets all outstanding technical requirements.

Kind regards,

Phuping Sucharitakul

Academic Editor

PLOS ONE
---

## [Editor Report · Acceptance letter]

PONE-D-25-22492R2

PLOS ONE

Dear Dr. Motta,

I'm pleased to inform you that your manuscript has been deemed suitable for publication in PLOS ONE. Congratulations! Your manuscript is now being handed over to our production team.

Kind regards,

on behalf of

Dr. Phuping Sucharitakul

Academic Editor

PLOS ONE